# Complete chemical structures of human mitochondrial tRNAs

Takeo Suzuki[1], Yuka Yashiro[1], Ittoku Kikuchi[1], Yuma Ishigami[1], Hironori Saito[2,3], Ikuya Matsuzawa[1], Shunpei Okada[1,4], Mari Mito[2], Shintaro Iwasaki [2,3], Ding Ma[1], Xuewei Zhao[1], Kana Asano[1], Huan Lin[1,5], Yohei Kirino [6], Yuriko Sakaguchi[1] & Tsutomu Suzuki [1✉]

Mitochondria generate most cellular energy via oxidative phosphorylation. Twenty-two species of mitochondrial (mt-)tRNAs encoded in mtDNA translate essential subunits of the respiratory chain complexes. mt-tRNAs contain post-transcriptional modifications introduced by nuclear-encoded tRNA-modifying enzymes. They are required for deciphering genetic code accurately, as well as stabilizing tRNA. Loss of tRNA modifications frequently results in severe pathological consequences. Here, we perform a comprehensive analysis of post-transcriptional modifications of all human mt-tRNAs, including 14 previously-uncharacterized species. In total, we find 18 kinds of RNA modifications at 137 positions (8.7% in 1575 nucleobases) in 22 species of human mt-tRNAs. An up-to-date list of 34 genes responsible for mt-tRNA modifications are provided. We identify two genes required for queuosine (Q) formation in mt-tRNAs. Our results provide insight into the molecular mechanisms underlying the decoding system and could help to elucidate the molecular pathogenesis of human mitochondrial diseases caused by aberrant tRNA modifications.

[1] Department of Chemistry and Biotechnology, Graduate School of Engineering, University of Tokyo, Bunkyo-ku, Tokyo 113-8656, Japan. [2] RNA System Biochemistry Laboratory, Cluster for Pioneering Research, RIKEN, 2-1 Hirosawa, Wako, Saitama 351-0198, Japan. [3] Department of Computational Biology and Medical Sciences, Graduate School of Frontier Sciences, University of Tokyo, Kashiwa, Chiba 277-8562, Japan. [4] Research Institute for Biomedical Sciences, Tokyo University of Science, 2669 Yamazaki, Noda, Chiba 278-0022, Japan. [5] State Key Laboratory of Marine Resource Utilization in South China Sea, Hainan University, 570228 Haikou, Hainan, P.R. China. [6] Computational Medicine Center, Sidney Kimmel Medical College, Thomas Jefferson University, Philadelphia, PA 19107, USA. ✉email: ts@chembio.t.u-tokyo.ac.jp

Mitochondria are eukaryotic organelles that generate chemical energy in the form of ATP through a process referred to as oxidative phosphorylation (OXPHOS)[1]. Hundreds to thousands of copies of mitochondrial (mt)DNAs are present within this organelle, implying that an endosymbiotic event took place when the ancestors of eukaryotes took in aerobic bacteria (in particular, Rickettsiales or close relatives) during early eukaryotic evolution. Human mtDNA is a circular and double-stranded DNA, 16,569 base pairs in length, which encodes 37 genes: 13 for the essential subunits of respiratory chain complexes I, III, IV, and V; 22 for tRNAs (mt-tRNAs); and two for rRNAs (mt-rRNAs)[2]. To translate those 13 mRNAs, mitochondria have their own protein synthesis machinery consisting of mitochondrial ribosomes (mitoribosomes), mt-tRNAs, and several translational factors[3]. The human mitoribosome contains ~80 ribosomal proteins as well as two mt-rRNAs (16S and 12S rRNAs) and mt-tRNA$^{Val}$ as structural components[4,5]. Thus, all RNA components required for mitochondrial translation are supplied by mitochondria themselves, whereas all the protein components, including ribosomal proteins, translational factors, aminoacyl-tRNA synthetases, and various factors required for processing and modification of RNAs, as well as biogenesis of mitoribosome, are encoded in the nuclear genome. They are translated in cytoplasm and imported to mitochondria.

In mammalian mitochondria, the decoding system is deviated from the canonical genetic code; AUA for Met, UGA for Trp, and AGR (R = A and G) for stop codons (see Table 1). The resultant 60 sense codons are deciphered by the 22 species of mt-tRNAs, which are the minimum set of tRNAs required for a decoding system in any organism and organelle[6]. To decrease the number of tRNAs required, all four codons in each family box are decoded by a single tRNA with unmodified uridine at the wobble position (position 34), as unmodified U34 can read any of four bases at the third position of a codon via a mechanism called four-way wobbling. On the other hand, two codon sets ending in purines are decoded by a tRNA with modified uridine at the wobble position, as this modification prevents misreading of pyrimidine-ending codons and stabilizes wobble pairing with G-ending codons. This is a general rule of the mitochondrial decoding system. In mammalian mitochondria, 5-taurinomethyluridine ($\tau$m$^5$U) is present at the wobble position in mt-tRNAs for Leu and Trp, and its 2-thiouridine derivative

($\tau$m$^5$s$^2$U) is present at the same position in the mt-tRNAs for Lys, Glu and Gln[7,8]. These modifications play critical roles in accurate decoding of cognate codons. In fact, structural analysis of the ribosomal small subunit revealed that $\tau$m$^5$U34-G3 pair uses Watson–Click geometry to stabilize the codon–anticodon interaction, possibly due to a favorable stacking interaction with the neighboring A35-U2 pair[9].

Disorders of human mitochondrial translation result in mitochondrial dysfunction with respiratory defects, and often cause severe clinical symptoms[6]. To date, 751 pathogenic mutations in mtDNA have been reported, of which 300 are in specific mt-tRNA genes (http://www.mitomap.org/MITOMAP)[10]. Pathogenic point mutations in mt-tRNAs are strongly associated with mitochondrial diseases[11,12]. We observed severe impairment of $\tau$m$^5$U modification in mutant tRNAs isolated from the cells of patients with mitochondrial encephalomyopathies[6,13]. mt-tRNA$^{Leu(UUR)}$-bearing pathogenic mutations including A3243G, U3271C, or other mutations associated with MELAS (mitochondrial myopathy, encephalopathy, lactic acidosis, and stroke-like episodes) lack $\tau$m$^5$U34 (refs. [14,15]), indicating that each of MELAS mutation hinders recognition by GTPBP3/MTO1, the enzymatic complex responsible for the $\tau$m$^5$U modification[16]. Biochemical studies revealed that the hypomodified mt-tRNA$^{Leu(UUR)}$ lacking $\tau$m$^5$U34 decodes the UUG codon less efficiently than the UUA codon[17]. In fact, mitochondrial ribosome profiling revealed that mitoribosomes stall at UUG codons in cybrid cells bearing A3243G mutation as well as in MTO1-knockout cells[18]. The hypomodified tRNA does not efficiently translate ND6, an essential component of complex I, because UUG codons are abundant in this gene[17]. This molecular mechanism explains the complex I deficiency observed in MELAS patients. Furthermore, the $\tau$m$^5$s$^2$U34 modification is not formed in mt-tRNA$^{Lys}$ with the A8344G mutation, which is associated with MERRF (myoclonus epilepsy associated with ragged red fibers)[19]. The hypomodified mt-tRNA$^{Lys}$ does not decipher its cognate codons efficiently[20], impairing mitochondrial protein synthesis and leading to defects in respiratory activity. These findings provided the first instances of human diseases caused by aberrant RNA modification, now referred to as RNA modopathy[16].

5-Formylcytidine (f$^5$C), another unique modification in mammalian mitochondria[21], is present at the wobble position of mt-tRNA$^{Met}$. f$^5$C34 is required to decipher the non-universal

**Table 1 Codon–anticodon pairing in the human mitochondrial genetic code.**

| Codon | Amino acid (anticodon) | Codon | Amino acid (anticodon) | Codon | Amino acid (anticodon) | Codon | Amino acid (anticodon) |
|---|---|---|---|---|---|---|---|
| UUU | Phe | UCU | | UAU | Tyr | UGU | Cys |
| UUC | (GAA) | UCC | Ser | UAC | (QUA) | UGC | (GCA) |
| UUA | Leu | UCA | (UGA) | UAA | Stop | **UGA** | Trp |
| UUG | ($\tau$m$^5$UAA) | UCG | | UAG | | UGG | ($\tau$m$^5$UCA) |
| CUU | | CCU | | CAU | His | CGU | |
| CUC | Leu | CCC | Pro | CAC | (QUG) | CGC | Arg |
| CUA | (UAG) | CCA | (UGG) | CAA | Gln | CGA | (UCG) |
| CUG | | CCG | | CAG | ($\tau$m$^5$s$^2$UUG) | CGG | |
| AUU | Ile | ACU | | AAU | Asn | AGU | Ser |
| AUC | (GAU) | ACC | Thr | AAC | (QUU) | AGC | (GCU) |
| **AUA** | Met | ACA | (UGU) | AAA | Lys | **AGA** | Stop |
| AUG | (f$^5$CAU) | ACG | | AAG | ($\tau$m$^5$s$^2$UUU) | **AGG** | |
| GUU | | GCU | | GAU | Asp | GGU | |
| GUC | Val | GCC | Ala | GAC | (QUC) | GGC | Gly |
| GUA | (UAC) | GCA | (UGC) | GAA | Glu | GGA | (UCC) |
| GUG | | GCG | | GAG | ($\tau$m$^5$s$^2$UUC) | GGG | |

Non-universal genetic codes are shown in bold: AUA for Met, UGA for Trp, and AGA/AGG for Stop. The anticodon sequences of each tRNA are indicated in parentheses.

AUA codon as Met as well as the canonical AUG codon[22,23]. We and others demonstrated that NSUN3 and ALKBH1 synthesize f⁵C34 (refs. [24–27]), and we identified two pathogenic point mutations in mt-tRNA^Met that impair f⁵C34 formation[24].

$N^6$-threonylcarbamoyladenosine (t⁶A) is a bulky modification found at position 37 (3′ adjacent to the anticodon) of tRNAs responsible for codons starting with A. t⁶A is a highly conserved and essential modification that plays critical roles in protein synthesis including aminoacylation, decoding, translocation, and so on. We showed that YRDC and OSGEPL1 are responsible for t⁶A37 formation, utilizing L-threonine, ATP, and $CO_2$/bicarbonate as substrates[28]. Curiously, t⁶A37 is dynamically regulated by sensing intracellular $CO_2$/bicarbonate concentration, suggesting that mitochondrial translation is modulated by t⁶A37 status[28]. We found 15 types of pathogenic mutations in mt-tRNA genes that impaired t⁶A37 formation, and confirmed that the levels of t⁶A37 were low in mt-tRNA^Thr bearing the A15923G mutation isolated from the cells of a patient with MERRF-like symptoms[28].

Early linkage analyses and recent genome-wide association studies with exome sequencing revealed pathogenic mutations in mt-tRNA-modifying enzymes associated with specific disorders; PUS1 in mitochondrial myopathy and sideroblastic anemia[29]; MTU1 in acute infantile liver failure[30–33]; GTPBP3 in hypertrophic cardiomyopathy, lactic acidosis, and encephalopathy or Leigh syndrome[34,35]; MTO1 in hypertrophic cardiomyopathy and lactic acidosis[36–38]; TRIT1 in encephalopathy and myoclonic epilepsy[39]; and NSUN3 in combined mitochondrial respiratory chain complex deficiency[27]. Genetic studies with mouse models revealed the significant roles of Cdk5rap1 in OXPHOS deficiency and cardiomyopathy[40], Mto1 in cardiomyocytes[41,42], and Mtu1 in liver failure[43].

To achieve a deeper understanding of human mt-tRNA modifications associated with molecular pathogenesis of human mitochondrial diseases, it is necessary to obtain a complete picture of post-transcriptional modifications in all 22 species of human mt-tRNAs. To date, only eight species of human mt-tRNAs have been sequenced and their post-transcriptional modifications determined. In a previous comprehensive analysis, we mapped 15 types of modified bases at 118 positions in all species of bovine mt-tRNAs[8], providing useful information for predicting human mt-tRNA modifications and their modifying enzymes. In this study, we isolate all 22 species of human mt-tRNAs and analyze the primary sequence and post-transcriptional modifications for each tRNA using RNA mass spectrometry in combination with biochemical analyses to map pseudouridines. In total, we identify 18 species of modified nucleosides at 137 positions in the 22 species of human mt-tRNAs. Here, we provide a complete list of mt-tRNA-modifying enzymes, which should facilitate investigations of their physiological roles.

## Results

### Isolation of all 22 species of human mt-tRNAs.
To date, eight species of human mt-tRNAs with post-transcriptional modifications have been analyzed: Asp[44], Ile[45], Lys[7,46], Leu(UUR)[7,47], Leu (CUN)[48], Pro[49], Ser(UCN)[50], and Ser(AGY)[51]. To conduct a comprehensive analysis of human mt-tRNA modifications, we used our established platforms for RNA isolation, chaplet column chromatography[52], and reciprocal circulating chromatography[53]. Using these techniques, we successfully isolated all 22 species of human mt-tRNAs from human placenta (Fig. 1) and several species of mt-tRNA from HeLa cells.

### Mass spectrometric analysis of human mt-tRNAs.
We conducted mass spectrometric analyses of 15 species of mt-tRNAs to

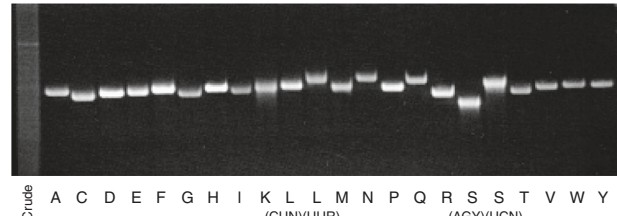

**Fig. 1 Isolation of human mitochondrial tRNAs.** Urea-denaturing gel electrophoresis of 22 species of human mt-tRNAs isolated from human placenta by chaplet column chromatography. The "crude" lane stands for placental total RNA. Each tRNA species is denoted by the single-letter abbreviation of the corresponding amino acid.

identify post-transcriptional modifications (Fig. 2, Supplementary Fig. 1, Supplementary Data 1). A typical example of modification mapping of human mt-tRNA^Cys (Fig. 2a) is shown in Fig. 2. First, we conducted LC/ESI-MS nucleoside analysis using hydrophilic chromatography (HILIC)[54] to detect modified nucleosides in mt-tRNA^Cys; the results revealed $N^6$-isopentenyladenosine (i⁶A), 1-methylguanosine (m¹G), pseudouridine (Ψ), and 1-methyladenosine (m¹A) (Fig. 2b). The presence of $N^6$-methyladenosine (m⁶A) (Fig. 2b) is probably due to spontaneous conversion from m¹A to m⁶A by the Dimroth rearrangement[55] during purification and handling of tRNA. Second, we used capillary LC/nano ESI-MS to analyze RNA fragments digested by RNase T₁ (Fig. 2c) and RNase A (Fig. 2d); this method revealed unmodified as well as modified fragments. To sequence and map the modified nucleosides, each fragment was further probed by collision-induced dissociation (CID). Unmodified fragments were unequivocally assigned to fragment sequences deduced from the reference mt-tRNA sequence by comparing the observed and calculated $m/z$ values (Fig. 2c, d, Supplementary Data 1). The internal fragments have 5′-hydroxyl (5′OH) and 3′-phosphate (3′p) groups, whereas pAGp and CUUCCA_OH were respectively assigned to the 5′- and 3′-terminal fragments of mt-tRNA^Cys (Fig. 2c, Supplementary Data 1) because mature tRNAs generally have 5′-monophosphate and 3′-hydroxyl groups. RNA fragments containing modified nucleosides can be assigned based on the difference between the observed mass and calculated $m/z$ value of the unmodified fragment. For instance, we found a species with $m/z$ 661.1, which corresponds to a doubly charged anion of 5′-AAACp-3′ with single methylation (Fig. 2e). CID analysis made it possible to deduce that the fragment sequence was 5′-AAACp-3′ in which the second A was methylated (Fig. 2f). Judging from the nucleoside analysis (Fig. 2b), it turned out to be m¹A58 in Am¹AACp (Fig. 2a, f). However, m¹A58 was a minor modification with 17% frequency (Fig. 2e). Likewise, m¹G and i⁶A were assigned at positions 9 and 37, respectively (Fig. 2a). According to the mass chromatograms (Fig. 2c, d), frequencies of m¹G9 and i⁶A37 are quite high with 100% and 99%, respectively (Supplementary Data 1 and 2).

Finally, we conducted whole-mass analysis of intact tRNA to validate our assignment. By deconvoluting multiply charged ions of mt-tRNA^Cys, we calculated the observed mass to be 22271.1 Da (Fig. 2g). Human mt-tRNA^Cys, which is composed of 69 nucleotides including m¹G, i⁶A and m¹A, was calculated to have a mass of 22286.0 Da. Given that m¹A is a minor modification and its frequency is quite low, the hypomodified tRNA was calculated to have a mass of 22272.0 Da which is consistent with the observed mass.

### Mapping of Ψ in human mt-tRNAs.
To analyze pseudouridine (Ψ), a mass-silent modification, we treated the tRNA with

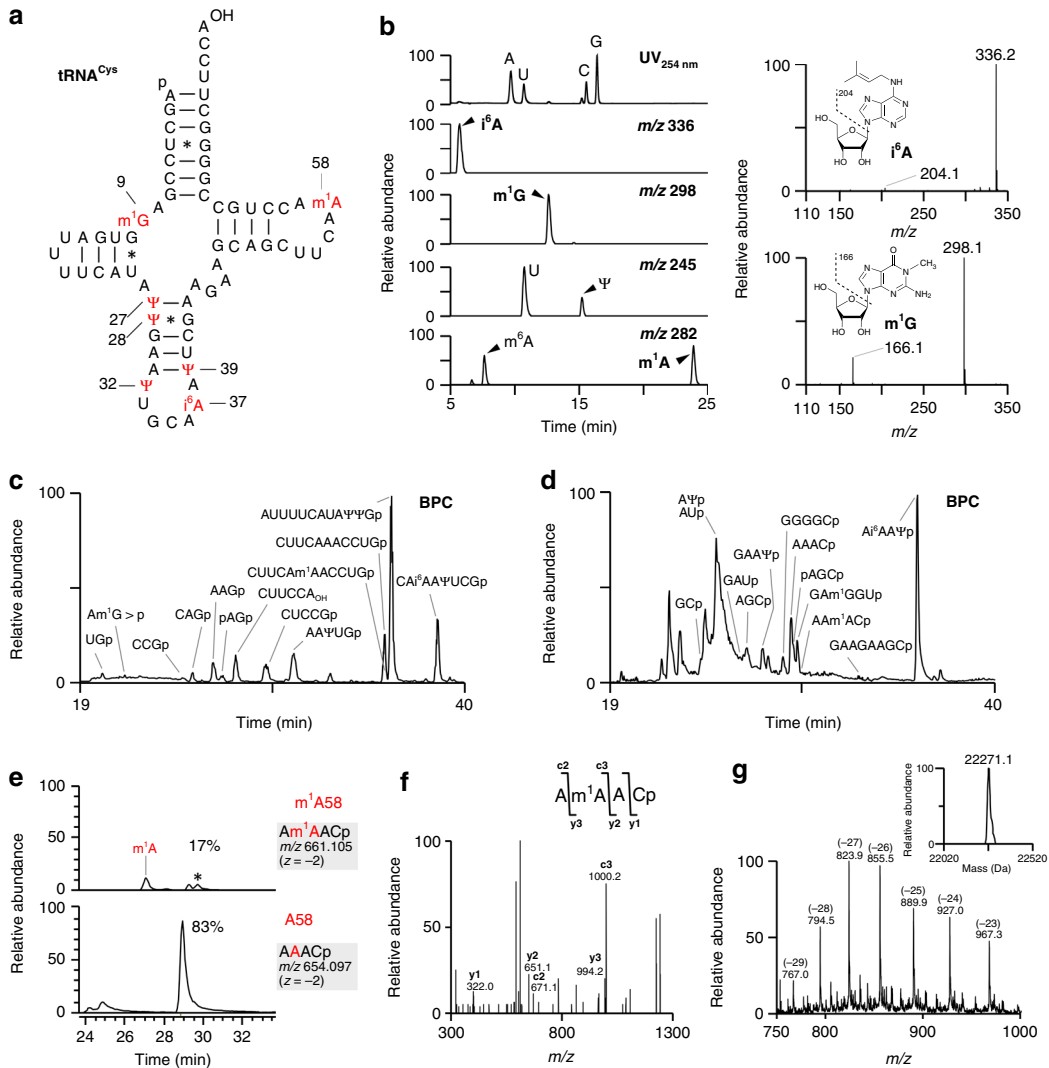

**Fig. 2 Mass spectrometric analyses of human mt-tRNA^Cys for assignment of post-transcriptional modifications. a** Secondary structure of mt-tRNA^Cys with modifications. "p" and "OH" stand for 5'-monophosphate and 3'-hydroxy termini, respectively. Symbols for modified nucleosides are as follows: m¹G 1-methylguanosine, Ψ pseudouridine, i⁶A N⁶-isopentenyladenosine, m¹A 1-methyladenosine. Watson–Crick and G–U pairs are indicated by solid lines and asterisks, respectively. **b** HILIC-MS nucleoside analysis of mt-tRNA^Cys. Left panels: UV chromatogram at 254 nm indicating four unmodified nucleosides (top panel), extracted-ion chromatograms (XIC) for the protonated ion of i⁶A (m/z 336) (second panel), m¹G (m/z 298) (third panel), Ψ (m/z 245) (fourth panel), and m¹A (m/z 282) (bottom panel). U and N⁶-methyladenosine (m⁶A) are also detected in the XIC of Ψ and m¹A, respectively. Right panels: Mass spectra of i⁶A (upper panel) and m¹G (lower panel). Dissociation of a base ion is indicated by a dotted line in each chemical structure. **c, d** RNA fragment analysis of mt-tRNA^Cys digested by RNase T₁ **c** and RNase A **d**. Assigned fragments (Supplementary Data 1) are indicated in the base peak chromatogram (BPC). p and >p represent the terminal phosphate and 2',3'-cyclic phosphate, respectively. **e** XICs of A58-containing fragments of mt-tRNA^Cys digested by RNase A with (upper panel) or without (lower panel) single methylation. The m¹A-containing fragment is indicated. The asterisk shows a fragment containing m⁶A which is produced from m¹A by the Dimroth reaction. The frequency of m¹A is calculated from the summed peak intensities of the m¹A- and m⁶A-containing fragments versus the non-methylated fragment. Sequence, m/z value, and charge state of each fragment are shown on the right. **f** Collision-induced dissociation (CID) spectrum of the RNase A-digested fragment of mt-tRNA^Cys. The doubly charged negative ion of the RNA fragment (m/z 661.1) was used as the precursor for CID. **g** Whole-mass analysis of intact mt-tRNA^Cys. A series of multiply charged negative ions with charge values are shown in the mass spectrum. The deconvoluted molecular mass is shown in the inset.

acrylonitrile to derivatize Ψ, allowing it to be detected by mass spectrometry (CE-Ψ-MS)[56]. As a result of this treatment, Ψ is cyanoethylated and converted to 1-cyanoethylpseudouridine (ce¹Ψ), which increases the mass by 53 Da. The cyanoethylated tRNA was digested by RNases and subjected to capillary LC/nano ESI-MS[57–59]. In human mt-tRNA^Cys, three cyanoethylated fragments were detected in the RNase T₁ digest (Supplementary Data 1), and each fragment was further analyzed by CID. As shown in Fig. 3a, ce¹Ψ was detected at position 39 by assigning the product ions of CAi⁶AAce¹ΨUCGp. In CE-Ψ-MS,

acrylonitrile could derivatize τm⁵(s²)U (Supplementary Fig. 2a, b) and queuosine (Q) (Supplementary Fig. 3a, b), in addition to ce¹Ψ, probably due to cyanoethylation of the secondary amino group of the side chain moieties of τm⁵(s²)U and Q by aza-Michael addition. We employed this method and analyzed Ψ's in the other 11 species of mt-tRNAs (Supplementary Data 3). The greatest advantage of CE-Ψ-MS is that it can detect Ψ in the 3' terminal region of RNAs, whereas no biochemical methods can detect Ψ in these regions. In fact, we identified two new Ψ's at positions 66 and 67 in human mt-tRNA^Pro by CE-Ψ-MS

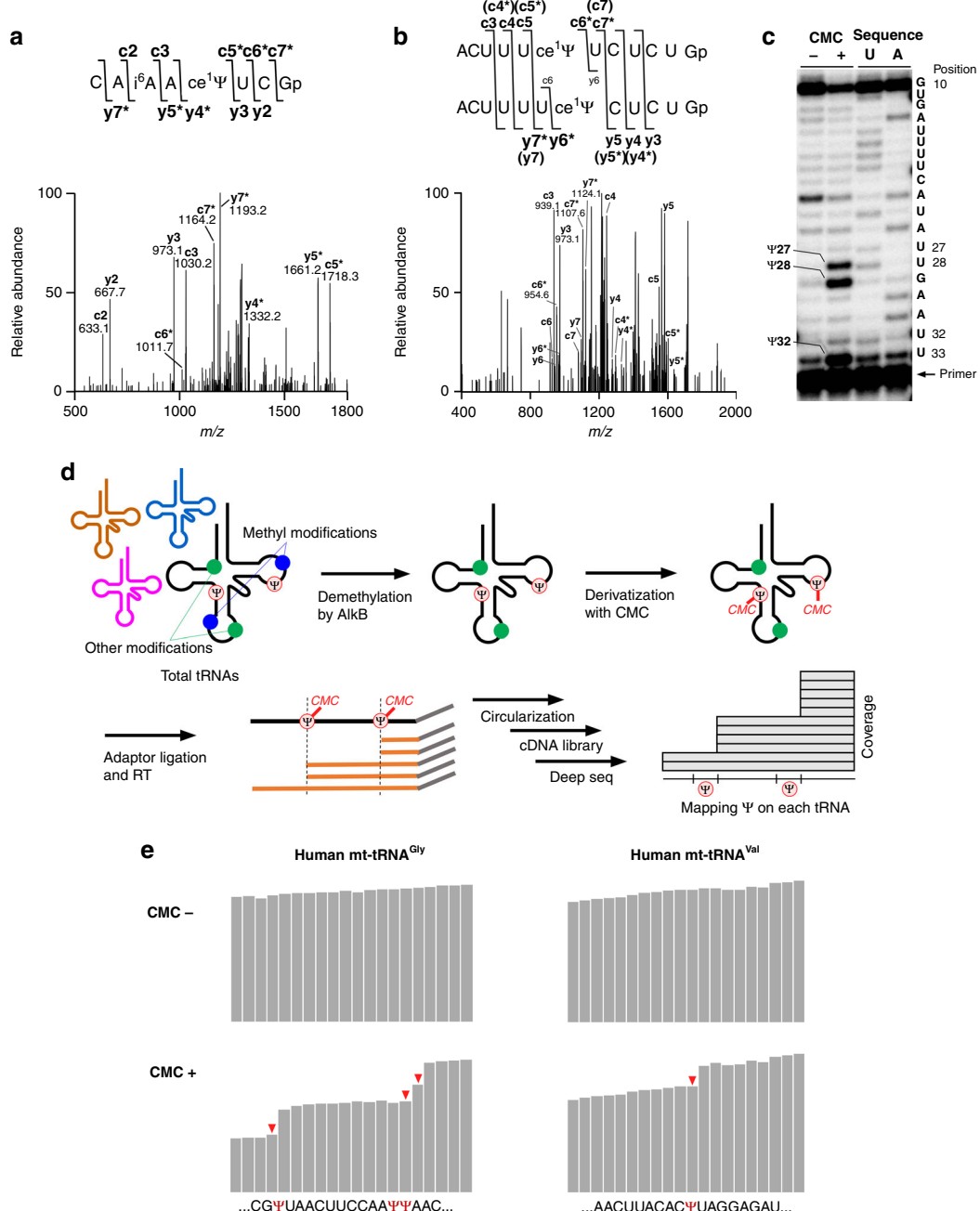

**Fig. 3 Mapping Ψ sites in human mt-tRNAs. a** A CID spectrum of the RNase $T_1$-digested fragment of human mt-tRNA$^{Cys}$. The doubly charged negative ion of the cyanoethylated RNA fragment (m/z 1345.7) was used as the precursor for CID. Asterisks in the spectrum denote product ions containing ce$^1$Ψ. **b** A CID spectrum of the cyanoethylated RNA fragment to determine the site for Ψs in human mt-tRNA$^{Pro}$. The triply charged negative ion of the mono-cyanoethylated RNA fragment (m/z 1266.5) was used as the precursor for CID. Assignment of the product ions revealed the presence of two fragments containing ce$^1$Ψ66 and ce$^1$Ψ67. Asterisks in the spectrum denote product ions containing ce$^1$Ψ. **c** CMC-PE analysis for detection of Ψ sites in human mt-tRNA$^{Cys}$. HeLa total RNA treated with (+) or without (−) CMC was reverse-transcribed with a primer specific to human mt-tRNA$^{Cys}$. The sequence ladders for U and A were generated under the same conditions in the presence of ddATP and ddTTP, respectively. The positions for Ψs and sequence are shown with the gel image. Source data are provided as a Source Data file. **d** Diagram of tRNA-Ψ-seq. Detailed procedure is described in the "Methods" section. **e** IGV snapshots of tRNA-Ψ-seq for human mt-tRNA$^{Gly}$ and mt-tRNA$^{Val}$. Histograms of mapped reads are shown for each tRNA when tRNA-Ψ-seq was performed in the absence (−) or presence (+) of CMC. The RT-stop signature for CMC-Ψ (indicated by red arrowheads) can be observed as a sudden drop in the number of piled reads.

(Fig. 3b), although the post-transcriptional modifications of this tRNA were reported previously[49]. We also detected Ψ at position 68 in human mt-tRNA$^{Ala}$ (Supplementary Data 3).

In addition to mass spectrometric analyses, we performed biochemical mapping of Ψ in mt-tRNAs. Ψ can be derivatized by the water-soluble carbodiimide CMC [N-cyclohexyl-N′-(2-

morpholinoethyl)carbodiimide methyl-p-toluenesulfonate]. $N^3$-derivatized Ψ (CMC-Ψ) can be detected by primer extension (CMC-PE)[60], as $N^3$-substitution hinders extension of cDNA when tRNA is reverse-transcribed. To perform the mapping, we treated total RNA from HeLa cells with CMC and used it as a template for primer extension. A specific region of each

individual tRNA was reverse-transcribed by a $^{32}$P-labeled primer. The radiolabeled cDNAs were resolved by PAGE analysis and visualized by a fluoroimager. As shown in Fig. 3c, three CMC-Ψ sites were clearly detected at positions 27, 28, and 32 of mt-tRNA$^{Cys}$ when we used CMC-treated total RNA as a template. We then applied CMC-PE to analyzed Ψs in the other eight species of mt-tRNAs (Supplementary Fig. 4, Supplementary Data 3). By combining the CE-Ψ-MS data and published information, we detected 52 Ψ sites in 22 species of human mt-tRNAs (Supplementary Data 3 and 4).

CMC-Ψs can be detected in a transcriptome-wide manner by deep sequencing[61]. We applied this technique to mapping Ψ in tRNAs (tRNA-Ψ-seq) (Fig. 3d). For this purpose, we treated human tRNA fraction with *Escherichia coli* AlkB to demethylate m$^1$A, 3-methylcytidine (m$^3$C), m$^1$G, $N^2$-methylguanosine (m$^2$G), and $N^2$, $N^2$-dimethylguanosine (m$^{2,2}$G) that prevent cDNA extension[62], followed by derivatizing Ψs in the resultant tRNAs with CMCT. cDNA libraries prepared with or without CMC treatment were subjected to deep sequencing, and sufficient data for both conditions were mapped to human mt-tRNA genes. The RT-stop signature for CMC-Ψ was calculated and designated the Ψ score (Supplementary Data 3). Typical examples of tRNA-Ψ-seq are shown in Fig. 3e. Because tRNA-Ψ-seq cannot detect Ψs in the 3′ terminal region of tRNA, three Ψ sites at positions 66–68 of mt-tRNAs for Ala and Pro were excluded in this analysis. Based on the scores, 44 of 49 sites were detected by tRNA-Ψ-seq, whereas five sites detected by CE-Ψ-MS and/or CMC-PE were not (Supplementary Data 3). The inconsistency in these results might be explained by differences between the cell lines in these experiments or by differences in detection efficiency.

**Full complement of RNA modifications in human mt-tRNAs.** In addition to mt-tRNA$^{Cys}$, we analyzed the other 13 mt-tRNAs whose modifications had not been previously reported (Fig. 4a, Supplementary Fig. 1, Supplementary Data 4). Of those, we analyzed mt-tRNA$^{Glu}$ by a conventional post-labeling method using two-dimensional thin-layer chromatography (2D-TLC) (Supplementary Fig. 5); eight modifications were clearly identified (Supplementary Data 4). Based on the peak intensities of XICs between the modified and unmodified RNA fragments, we calculated modification frequencies for 51 sites in 15 species of mt-tRNAs (Supplementary Data 1 and 2).

m1A9 and m2G10 are quite abundant modifications found in eight mt-tRNA species. Their frequencies are high in six species of them, whereas hypomethylation was found in Phe and His.

Exclusively in mt-tRNA$^{Arg}$, m$^1$A is present at position 16 in the D-loop. Judging from mass chromatographic peak ratio between modified and unmodified fragments, we estimated the modification frequency of m$^1$A16 to be about 20% (Fig. 5a). Partially modification of m$^1$A16 in this tRNA was reported previously[63].

In mammalian mitochondria, τm$^5$U is present at the wobble position (position 34) in mt-tRNAs for Leu(UUR) and Trp, whereas τm$^5$s$^2$U is present at the corresponding positions in mt-tRNAs for Lys, Glu, and Gln[6,8]. In human mitochondria, τm$^5$U and τm$^5$s$^2$U are present at position 34 of the mt-tRNAs for Leu (UUR) and Lys, respectively[7]. In this study, we confirmed the presence of τm$^5$U34 in human mt-tRNA$^{Trp}$ with 79% frequency (Fig. 5b), and τm$^5$s$^2$U34 in the human mt-tRNAs for Glu (Supplementary Fig. 5) and Gln (Fig. 5c). Modification status of the wobble position of mt-tRNA$^{Gln}$ is quite complex with τm$^5$s$^2$U34 (67%), τm$^5$U34 (12%), s$^2$U34 (11%), and U34 (8%) (Supplementary Data 2). In addition, we detected a small amount of 5-carboxymethylaminomethyluridine (cmnm$^5$U, 0.6%) in mt-tRNA$^{Trp}$ (Fig. 5b), and carboxymethylaminomethyl-2-thiouridine

(cmnm$^5$s$^2$U, 1.3%) in mt-tRNA$^{Gln}$ (Fig. 5c). We previously reported that cmnm$^5$U and cmnm$^5$s$^2$U are observed in mt-tRNAs under depletion of taurine[16]. cmnm$^5$(s$^2$)U are naturally present in these tRNAs under normal growth condition.

According to previous studies[8], Q is present at position 34 of mammalian mt-tRNAs for Tyr, His, Asp, and Asn. In addition to Q34 in human mt-tRNA$^{Asp44}$, we detected Q34 in the other three tRNAs.

We assumed that human mt-tRNA$^{Met}$ contains 5-formylcytidine (f$^5$C) and two Ψs, based on bovine mt-tRNA$^{Met}$ sequence[21,64] (Spremulli, L. L., personal communication). In addition to Ψ27, Ψ50, and f$^5$C34, which were predicted, we detected Ψ55 in this tRNA. As we reported previously[24], f$^5$C34 is fully introduced to human mt-tRNA$^{Met}$ (Supplementary Data 2).

2-Methylthio-$N^6$-isopentenyladenosine (ms$^2$i$^6$A) is present at position 37 in mt-tRNAs for Phe, Ser(UCN), Trp, and Tyr. Although the frequency of this modification is quite high in Phe (93%) (Supplementary Data 2), it is partially introduced to Trp (ms$^2$i$^6$A = 51%, i$^6$A = 31%) (Fig. 5b, Supplementary Data 2) and Tyr (ms$^2$i$^6$A = 45%, i$^6$A = 17%) (Fig. 5d, Supplementary Data 2).

t$^6$A is another critical modification at position 37 of mt-tRNAs for Ile, Lys, Asn, Ser(AGY), and Thr. It is fully introduced to Thr (97%) and partially to Asn (42%) (Fig. 5e) and Ser(AGY) (74%) (Supplementary Data 2).

m$^1$G is present at position 37 of mt-tRNAs for Ala, Leu(CUN), Pro, and Gln. It is completely introduced to Ala, Pro, and Gln (Supplementary Data 2).

5-Methyluridine (m$^5$U), so-called ribothymidine, is present at position 54 of five mt-tRNAs. Since bovine mt-tRNAs do not have this modification[8], m$^5$U54 is a modification unique to human mt-tRNAs. Rate of m$^5$U54 is quite high in mt-tRNAs for Pro (98%) and Asn (95%), but low in mt-tRNA$^{Gln}$ (33%) (Fig. 5f, Supplementary Data 2).

5-Methylcytidines (m$^5$C) is present in six species of human mt-tRNAs[27,65,66]. The rate of m$^5$C is quite high in Phe (91%), His (100%), and Tyr (89%) (Supplementary Data 2). Three consecutive m$^5$C are present at positions 48–50 in mt-tRNA$^{Ser(AGY)}$. About 80% of this tRNA has all three m$^5$Cs, whereas the rest 20% has two of them (Supplementary Data 2).

In mt-tRNA$^{His}$, we detected two kinds of RNase T$_1$-digested fragments with different residues at position 47 (A47 and U47) (Supplementary Fig. 1, Supplementary Data 1). According to the database of polymorphilic mutations in human mtDNA, this variation is derived from A12180T mutation in a type of haplogroup A2 [GenBank accession number KJ923817.1 (ref. [67])], suggesting that this heteroplasmy for this mutation is harmless. Intriguingly, the A47 variant has m$^5$C48 (100%), whereas the U47 variant has no m$^5$C48 (Supplementary Data 1). In addition, we also detected addition of G$_{-1}$ to the 5′ terminus of this tRNA mediated by THG1L[68].

**Biogenesis of Q34 in human mt-tRNAs.** In human cytoplasm, Q34 is present in several cytoplasmic tRNAs responsible for NAY codons. Q34 is synthesized by tRNA guanine transglycosylase (TGT), which catalyzes substitution of guanine with queuine, a nucleobase of Q34 (ref. [69]). Unlike the bacterial TGT which is a single protein, eukaryotic TGT consists of a catalytic subunit QTRT1 and a non-catalytic subunit QTRT2 (formerly known as QTRTD1)[70]. Both QTRT1 and QTRT2 are required for Q34 formation on cytoplasmic tRNAs. We here identified Q34 in four species of human mt-tRNAs. To examine whether QTRT1 and QTRT2 are also required for Q34 biogenesis of mt-tRNAs, we knocked out *QTRT1* and *QTRT2* in HEK293T cells by CRISPR/Cas9 system (Fig. 6a). We then isolated the mt-tRNAs for Asp, His, Asn, and Tyr from WT and KO cells, and subjected them to

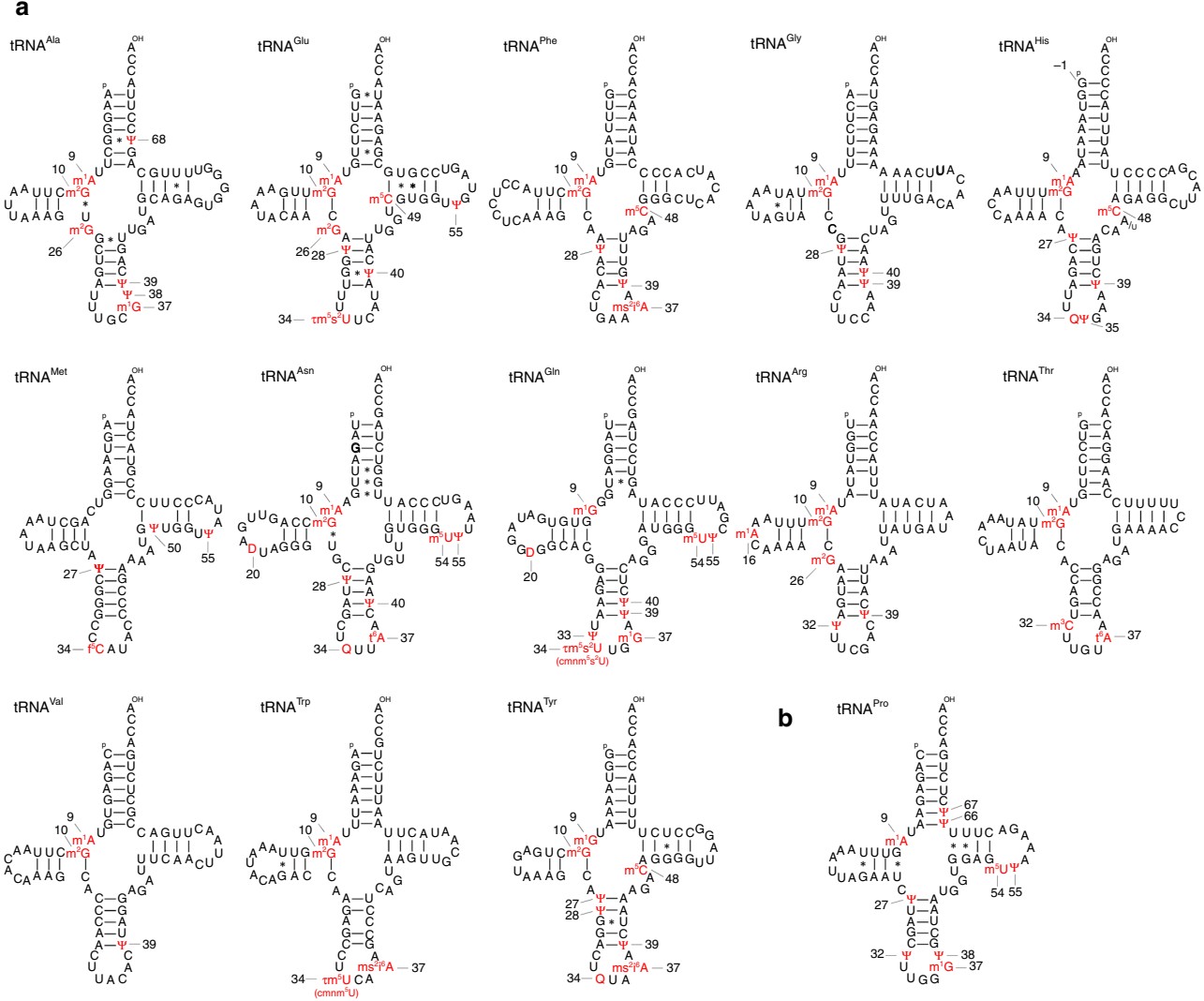

**Fig. 4 Post-transcriptional modifications in 14 human mt-tRNAs. a** Secondary structure of 13 human mt-tRNAs with post-transcriptional modifications. Symbols for modified nucleosides are as follows: m2G $N^2$-methylguanosine, m5C 5-methylcytidine, ms2i6A 2-methylthio-$N^6$-isopentenyladenosine, D dihydrouridine, m5U 5-methyluridine. The A/U in mt-tRNA$^{His}$ indicates that both A and U were detectable at this position as a result of heteroplasmy in the sample. **b** Secondary structure of human mt-tRNA$^{Pro}$, including unreported Ψ at positions 66 and 67.

LC/MS nucleoside analysis. We detected no Q signal in any of mt-tRNA isolated from cells harboring knockout of *QTRT1* or *QTRT2* (Fig. 6b), demonstrating that both genes are required for Q34 biogenesis of mt-tRNAs.

To pursue functional relevance of this modification, we have carried out mitoribosome profiling of *QTRT2* KO versus wild-type HEK293T cells, and calculated codon occupancy change upon *QTRT2* KO. As shown in Fig. 6c, the codon occupancy at UAU codon increased significantly upon *QTRT2* KO, indicating that Q34 in mt-tRNA$^{Tyr}$ plays a role in efficient decoding of UAU codon in mitochondria. Dysregulation of cytoplasmic translation upon queuine depletion results in unfolded proteins that trigger endoplasmic reticulum stress and activation of the unfolded protein response in human cell lines[71]. To characterize cellular protein homeostasis in *QTRT2* KO, we measured the formation of protein aggregates using destabilized variants (R188Q and R188Q/R261Q) of firefly luciferase (Fluc) fused with EGFP[72]. The EGFP-tagged Fluc variant is a sensitive reporter to measure the formation of protein aggregates. As shown in Fig. 6d, EGFP foci due to folding problem of Fluc variants significantly increased in *QTRT2* KO compared with WT in both reporters with single and

double mutations. Even in the reporter without mutation, the aggregation rate of *QTRT2* KO was highly than that of WT. This result strongly suggests that a protein folding problem takes place in *QTRT2* KO due to lack of Q34 in both cytoplasmic and mitochondrial tRNAs.

## Discussion

Here, we report the post-transcriptional modifications in 14 species of human mt-tRNAs. Together with previously published information for the eight mt-tRNAs, somewhat revised on the basis of our findings, we now have a complete picture of post-transcriptional modifications of human mt-tRNAs (Supplementary Data 4), comprising 18 types of RNA modifications at 137 positions (8.7% of 1575 total residues) (Fig. 7). All modifications are summarized in cloverleaf structures of 22 human mt-tRNAs (Supplementary Fig. 6). Notably, the absence of 2′-O-methylation is a characteristic feature of mammalian mt-tRNAs. The numbers of modified sites range from three to nine, corresponding to 4.2–12.7% of tRNA residues. Human mt-tRNA$^{Val}$ is the least modified tRNA, bearing only three sites, m1A9, m2G10, and Ψ39,

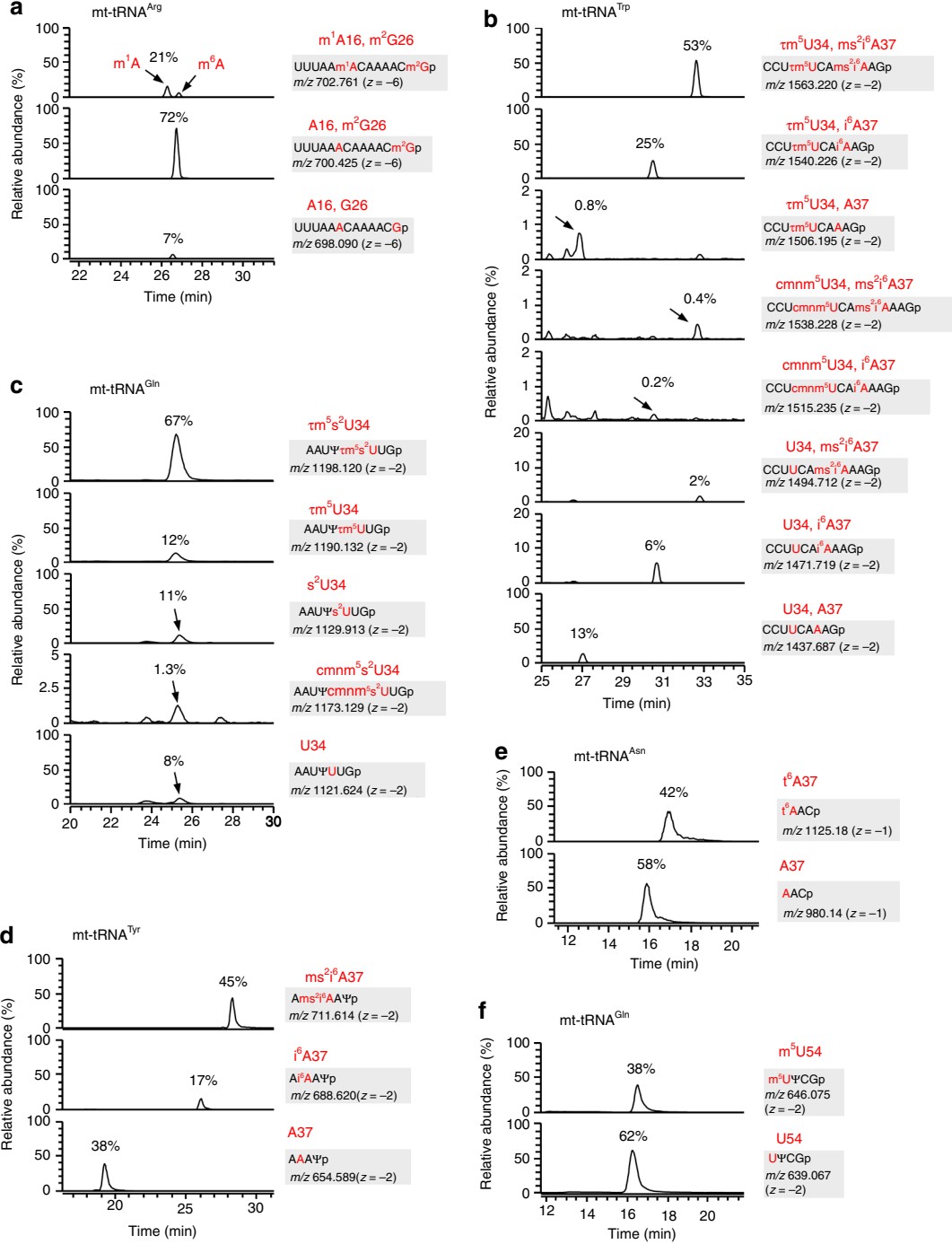

**Fig. 5 Frequencies of mt-tRNA modifications.** Typical XICs of modified and unmodified RNA fragments of the indicated tRNAs digested by RNase T1 (**a–c**, **f**) or RNase A (**d**, **e**). The frequency of each modification is calculated from peak intensity ratio of the modified and unmodified RNA fragments. Sequence, m/z value, and charge state of each fragment are shown on the right.

whereas mt-tRNAs for Leu(UUR), Asn, Pro, and Gln have the highest number of modified sites (nine sites).

We previously reported 15 types of RNA modifications at 118 positions in bovine mt-tRNAs[8]. The larger number of modifications and modified positions in human mt-tRNAs can be attributed to the presence of m5U54 and cmnm5(s2)U34 (minor). Because the 22 species of mt-tRNAs share 81% sequence identity between human and bovine, it is intriguing to compare tRNA modifications between the two species. Ninety-seven positions with modifications are common to both species, whereas 40 and 22 positions are specific to human and bovine mt-tRNAs,

respectively. Seventeen of 40 human-specific positions are not the same residues with those in bovine mt-tRNAs, indicating natural loss of modifications by mutations, and conversely, 14 out of 22 are bovine-specific residues. These facts indicate that the modification system for mt-tRNAs is well-conserved between these two mammals. Presumably, some alteration in tRNA sequences between these species modulates positive and negative determinants for each tRNA-modifying enzyme, resulting in differences in modification status between the species.

In regard to the decoding system of human mitochondria, we confirmed that eight mt-tRNAs responsible for family boxes had

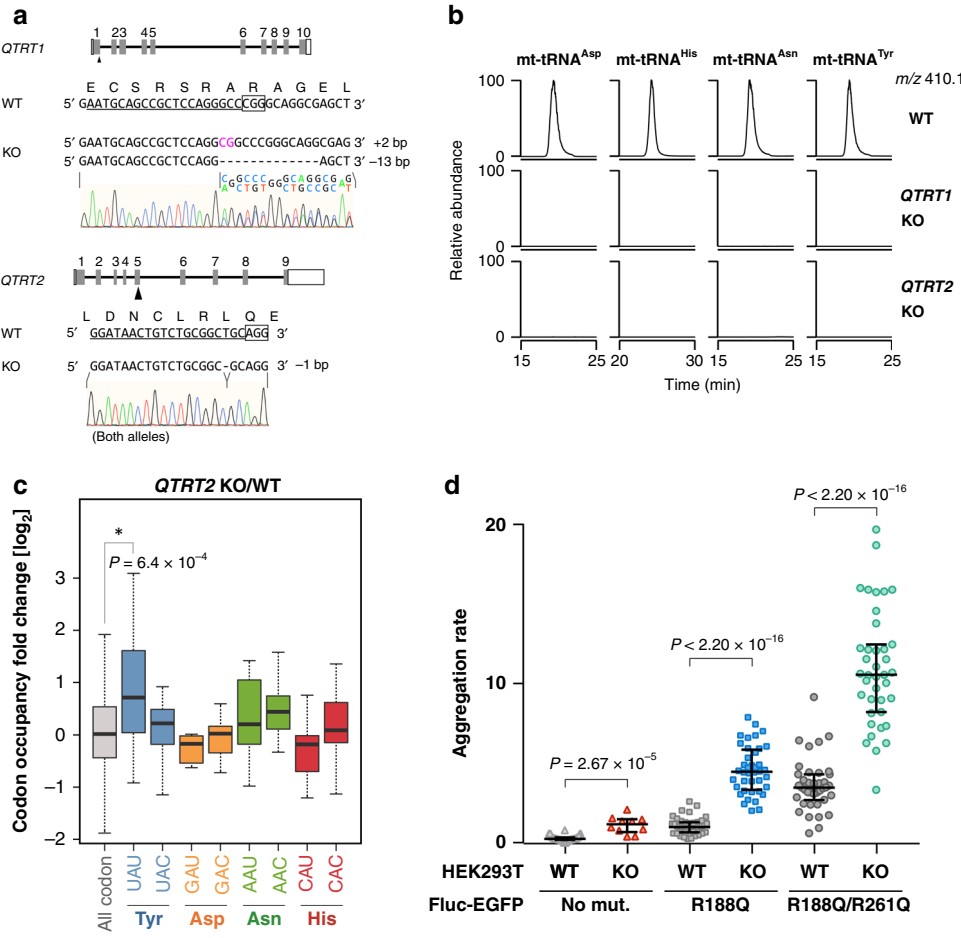

**Fig. 6 QTRT1 and QTRT2 are necessary for Q34 in mt-tRNAs. a** Gene structure of the human *QTRT1* (upper) and *QTRT2* (lower) with mutation sites introduced by the CRISPR/Cas9 system. The target sequence of the single-guide RNA (sgRNA) is underlined. The protospacer adjacent motif (PAM) sequence is boxed. Insertion and deletion are indicated by magenta letters and dashed lines, respectively. Electropherogram of Sanger sequence is shown in each KO cell. **b** XICs of Q nucleoside of human mt-tRNAs for Asp, His, Asn, and Tyr isolated from WT (top), *QTRT1* KO (middle), and *QTRT2* KO cells (bottom). **c** Boxplot of fold change in the ribosomal A-site codon occupancy of mitoribosome from *QTRT2* KO versus WT HEK293T cells. Codon occupancy is calculated by average of biological replicates ($n = 2$). Each box shows the first quartile, median, and third quartile. The whiskers represent the 1.5× interquartile ranges. *$P < 1.0 \times 10^{-3}$ value was calculated by two-sided Wilcoxon rank-sum test. **d** Proteome stress of *QTRT2* KO measured by the firefly luciferase (Fluc)-based thermal stability sensor fused with enhanced green fluorescent protein (EGFP). The Fluc sensor with no mutation (No mut.), single-(R188Q) or double-(R188Q/R261Q) mutation was introduced to *QTRT2* KO or WT HEK293T cells. The aggregation rate was calculated by dividing the number of sensor protein aggregates by the area of EGFP fluorescence in each microscopic images. Horizontal lines in the scattered dot plot represent the first quartile, median, and third quartile from the bottom. *P* value were calculated by two-sided Wilcoxon rank-sum test. Source data are provided as a Source Data file.

unmodified U34 in the anticodon (Table 1), demonstrating that the family boxes are decoded by single tRNAs via the four-way wobble rule. For NNR codons, all six mt-tRNAs had modifications at the wobble position: f$^5$C34 in mt-tRNA$^{Met}$, τm$^5$U34 in mt-tRNAs for Leu(UUR) and Trp, and τm$^5$s$^2$U in mt-tRNAs for Lys, Glu, and Gln. These modifications contribute to precise and efficient recognition of the cognate codons, as well as prevention of near-cognate codon misreading. Q34 is present in four mt-tRNAs (Tyr, His, Asn, and Asp) responsible for NAY codons.

Given that pathogenic mutations related to mitochondrial dysfunction are frequently found in tRNA-modifying enzymes, a complete list of human genes responsible for mt-tRNA modifications is necessary to fully appreciate the physiological and pathological importance of mt-tRNA modifications. In addition, identification of tRNA-modifying enzymes enables us to study the biosynthetic pathway responsible for each modification. Based on the finding in this study, we provide a list of confirmed and predicted genes responsible for mt-tRNA modifications,

accompanied by information about the human diseases associated with each gene (Supplementary Table 1). About 30 genes are responsible for tRNA modifications in human mitochondria.

m$^5$U54 is present in human mt-tRNAs but not in bovine mt-tRNAs. This is the only modification unique to one of these species. Very recent works showed that *TRMT2B*, a mammalian homolog of bacterial TrmA, is responsible for m$^5$U54 in several mt-tRNA species, as well as for m$^5$U429 in 12 S rRNA[73,74]. TrmA has a catalytic Cys residue which is essential to m$^5$U54 formation[75]. Human and mouse *TRMT2B* conserve this Cys residue, whereas bovine *TRMT2B* homolog has a Cys-to-Tyr point mutation at the corresponding site that inactivates the enzyme.

m$^3$C32 is present in mt-tRNAs for Ser(UCN) and Thr. We and others previously reported that yeast ABP140/Trm140 is responsible for m$^3$C32 formation in yeast cytoplasmic tRNAs[76,77]. In humans, there are four paralogs of yeast ABP140/Trm140: METTL2A, METTL2B, METTL6, and METTL8. The first three synthesize m$^3$C of cytoplasmic tRNAs, whereas

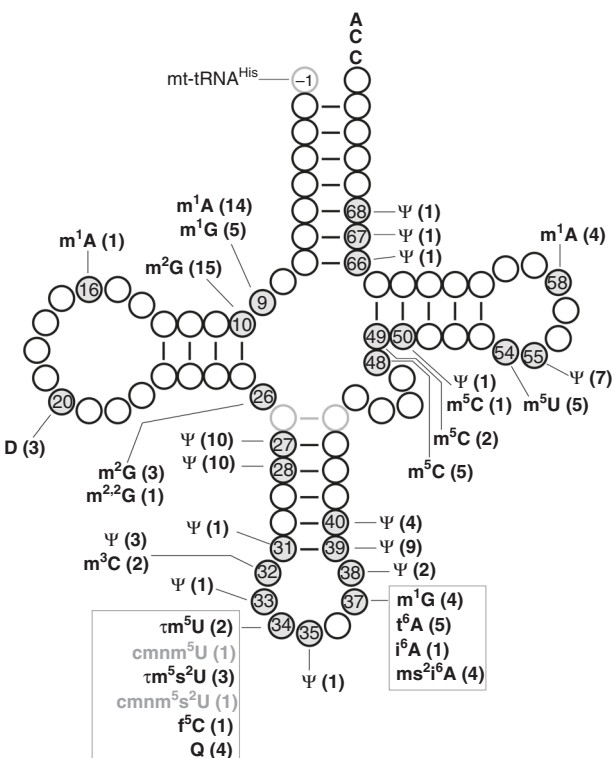

**Fig. 7 Integrated view of post-transcriptional modifications in 22 human mt-tRNAs.** The positions of modifications are indicated by gray circles on the schematic cloverleaf structure of tRNA with position numbers. At each position, the symbol of each modification is shown with the number of tRNA species in parenthesis. The light gray circles and line represent a base pair between positions 27a and 43a, which are unique to mt-tRNA$^{Ser(UCN)}$. The "−1" for G$_{-1}$ is specific to mt-tRNA$^{His}$. The symbol m$^{2,2}$G represents N$^2$, N$^2$-dimethylguanosine.

METTL8 is responsible for mRNA modification[78]. It remains to be determined which paralog is responsible for m$^3$C32 formation in mt-tRNAs.

Q34, a unique modification with 7-deazaguanosine structure, is present in both cytoplasmic and mitochondrial tRNAs responsible for NAY codons. Q34 controls the elongation speed of cognate codons, thereby modulating correct folding of nascent proteins[71]. Q34 is present in four mt-tRNAs for Tyr, His, Asn, and Asp. In this study, we found that both *QTRT1* and *QTRT2* are essential for Q34 formation in four mt-tRNAs. The physiological impact of queuosine depletion[71,79] could be attributed to hypomodification of Q34 in mt-tRNAs, as well as in cytoplasmic tRNAs. We here carried out mitoribosome profiling of *QTRT2* KO versus wild-type HEK293T cells, and calculated codon occupancy change upon *QTRT2* KO. The occupancy at UAU codon increased significantly upon *QTRT2* KO (Fig. 6c), indicating that Q34 in mt-tRNA$^{Tyr}$ plays a role in efficient decoding of UAU codon in mitochondria. In addition, we found a protein folding problem due to lack of codon optimality in *QTRT2* KO cells (Fig. 6d). However, it is difficult to estimate the effect of Q34 deficiency in mt-tRNAs on the cytoplasmic protein homeostasis. Our group previously reported that τm$^5$U34 deficiency in mt-tRNAs induced protein aggregation in cytoplasm[42]. This fact naturally speculates that Q34 deficiency in mt-tRNAs could affect cytoplasmic protein homeostasis to some extent. Recently, it has been shown that queuine depletion promotes a Warburg type metabolism with increased glycolysis and glutaminolysis with increased ammonia and lactate production, causing a decreased rate of ATP synthesis[80]. However, no proliferation phenotype was not observed upon queuine depletion[80]. Mitochondrial Q34 deficiency could slightly modulate mitochondrial translation and affect cellular metabolic status.

m$^1$A16 is unique to human mt-tRNA$^{Arg}$. Two methyltransferases are responsible for m$^1$A formation in mammalian mitochondria. TRMT61B introduces m$^1$A58 into several mt-tRNAs[81] and m$^1$A947 in 16S rRNA[82]. Judging from the consensus loop sequence recognized by TRMT61B, m$^1$A16 might also be introduced by TRMT61B. On the other hand, m$^1$A9 and m$^1$G9 in a subset of mt-tRNAs are introduced by the MRPP1 (TRMT10C) and MRPP2 (SDR5C1) complex[83].

Ψ is the most abundant modification in human mt-tRNAs[84]. Because Ψ is a mass-silent modification, it is necessary to derivatize Ψ with chemical modification, such as cyanoethylation or CMC addition, in order to detect it. However, because the frequency of Ψ varies at each position and most sites are partially modified, it is extremely difficult to detect minor sites with low Ψ frequencies. In addition, the level of Ψ might differ among cell types and tissues. Moreover, chemical derivatization of Ψ is strongly affected by tRNA structure, making it even more difficult to determine and compare the exact frequency of Ψ at different positions. In this study, we employed three techniques for detecting Ψ, CE-Ψ-MS, CMC-PE, and tRNA-Ψ-seq, and carefully mapped Ψ at 52 positions in human mt-tRNAs. Ψ33, Ψ35, Ψ38, Ψ66, and Ψ68 are present in human, but not in bovine, mt-tRNAs. Ψ is introduced by isomerization of uridine catalyzed by Ψ synthase. The human genome encodes 13Ψ synthases[85]. In this study, we identified Ψ35 in human mt-tRNA$^{His}$, the first time this modification has been detected in a mammalian mt-tRNA. Although Ψ35 was previously identified in starfish mt-tRNA$^{Asn}$ and eukaryotic cytoplasmic tRNA$^{Tyr}$[86,87], the responsible Ψ synthase has not yet been identified. Given that yeast Pus7p has broad substrate specificity and introduces Ψ into U2 snRNA, 5S rRNA, and cytoplasmic tRNAs, including Ψ35 (ref. [88]), human PUS7 might be involved in formation of Ψ35 in mt-tRNA$^{His}$. Further study is necessary to confirm this prediction. Similarly, based on the fact that yeast Pus3p introduces Ψ38 and Ψ39 in cytoplasmic tRNAs, Ψ38 might be introduced by human PUS3. Human PUS1 is a plausible candidate responsible for Ψ66 and Ψ68, as Ψ67 is introduced by this enzyme[89]. Regarding Ψ33, four Ψ synthases are predicted as candidates.

We identified m$^5$Cs at positions 48–50 in six mt-tRNA species. Recently, we reported that a fraction of NSUN2 is localized in mitochondria and introduces m$^5$C in mt-tRNAs[66,90].

MTU1 and NFS1 are responsible factors for 2-thiouridine formation of τm$^5$s$^2$U34 (ref. [91]). However, in *E. coli* and other bacteria[92], five sulfur mediators TusA, B, C, D, and E are required for 2-thiouridine formation in addition to IscS and MnmA. IscS, an NFS1 homolog in bacteria, catalyzes desulfuration of L-cysteine, and transfers a persulfide sulfur to TusA. Then, the sulfur is transferred to TusD in TusBCD complex, and to TusE. Finally, TusE interacts with MnmA-tRNA complex and the sulfur is transferred to the wobble uridine in tRNA mediated by MnmA. Based on the bacterial studies, the sulfur relay system might also be involved in mitochondrial 2-thiouridine formation. Some sulfur mediators could be added to the list of genes.

m$^2$G6 and m$^5$C72 are present in bovine mt-tRNA$^{Asp}$ and mt-tRNA$^{Thr}$[8], respectively, but absent from human mt-tRNAs. It is likely that A6 in human mt-tRNA$^{Asp}$ simply avoids m$^2$G6 modification. On the other hand, C72 is retained in human mt-tRNA$^{Thr}$. Some human cytoplasmic tRNAs also have m$^5$C72 formed by NSUN6 methyltransferase[93]. If bovine NSUN6 has a weak mitochondria-targeting sequence, the enzyme might be partially localized to mitochondria, where it introduces m$^5$C72 in mt-tRNA$^{Thr}$, whereas human NSUN6 might not be imported into mitochondria at all. Otherwise, a structural difference

between human and bovine mt-tRNAs$^{Thr}$ might be responsible for the species-specific m$^5$C72 modification.

Pathogenic point mutations in mt-tRNA genes are associated with human mitochondrial diseases[11]. We previously showed that several mutations in mt-tRNAs associated with MELAS and MERRF prevent recognition by tRNA-modifying enzymes responsible for τm$^5$s$^2$U34 formation, resulting in loss of the wobble modifications[6,13]. This finding indicates that some pathogenic point mutations act as negative determinants for tRNA-modifying enzymes. In addition, if such a point mutation overlaps with the site to be modified, it would likely eliminate the modification. Of ~300 tRNA mutations associated with human diseases in the MITOMAP[10], 23 pathogenic point mutations in 13 species of mt-tRNAs can be found in modification sites (Supplementary Data 5), many of which are present in the tRNA body, implying that loss of tRNA modification as a result of pathogenic mutation is a primary cause of mitochondrial disease.

In summary, we have provided a complete list of post-transcriptional modifications in human mt-tRNAs. This information represents a milestone in the study of biological aspects of tRNA modifications in human mitochondria. Identification of mt-tRNA-modifying enzymes enables us to investigate and study the molecular pathogenesis of RNA modopathies associated with mitochondrial dysfunction. Recent studies have shown that tRNA modifications are dynamically regulated, and that their frequencies change under various cellular conditions[16,28]. However, the currently available methods for analyzing RNA modification have both advantages and disadvantages. RNA-MS is an ideal and useful technique for quantifying the exact frequency of any tRNA modifications, except for Ψ. For this analysis, however, we need to isolate each tRNA species from various cells and tissues. On the other hand, epitranscriptome-wide methods based on deep sequencing technology can be used to quantify a couple of tRNA modifications in a limited quantity of specimens from various sources. In the future, we need to develop an innovative and sensitive technology for quantifying any tRNA modifications more easily and accurately.

## Methods

**Isolation of individual mt-tRNAs from human placenta and HeLa cells**. Preparation of human placental RNA was performed in the literature[17]. Briefly, total RNA was extracted from buffer-homogenized human placenta by phenol extraction, and the tRNA fraction was roughly fractionated by anion exchange chromatography with DEAE Sepharose Fast Flow (GE Healthcare). Twenty-two species of individual mt-tRNAs were isolated by chaplet column chromatography[52]. Total RNA was extracted from HeLa cells using TriPure Isolation Reagent (Roche Life Science). Several mt-tRNA species were isolated by reciprocal circulating chromatography method[53]. The isolated tRNAs were resolved by denaturing polyacrylamide gel electrophoresis with 7 M urea and gel-purified. The sequences of DNA probes for mt-tRNA isolation are listed in Supplementary Data 6.

**RNA mass spectrometry**. For nucleoside analysis of tRNAs[54], the isolated tRNA (10 pmol) was digested into nucleosides 5′-monophosphate with 0.2 U nuclease P1 (FUJIFILM Wako Pure Chemical) in 10 mM ammonium acetate (pH 5.3) at 37 °C for 60 min, followed by adding 0.1 U phosphodiesterase I (Worthington) in 50 mM ammonium bicarbonate (to adjust pH from 5.3 to 8.2) and incubating at 37 °C for 60 min. The digests were dephosphorylated by adding 0.1 U of bacterial alkaline phosphatase (from *E. coli* C75; FUJIFILM Wako Pure Chemical) and incubating at 37 °C for 60 min. The digested nucleosides were dissolved in 90% acetonitrile/10% water, and subjected to a hydrophilic interaction LC (ZIC-cHILIC, 3 μm particle size, 2.1 × 150 mm; Merck) connected to ESI-MS on a Q Exactive hybrid Quadrupole-Orbitrap mass spectrometer (Thermo Fisher Scientific), equipped with an ESI source and an Ultimate 3000 liquid chromatography system (Thermo Fisher Scientific). The nucleosides were separated at a flow rate of 100 μL/min in a solvent system consisting of 5 mM ammonium acetate (pH 5.3) (solvent A) and acetonitrile (solvent B) in a multistep linear gradient (90–40% B from 0 to 30 min, 40% B for 10 min, and then initialized to 90% B)[54]. For queuosine analysis, a linear gradient was set to starting with 50% B in 0–5 min, 50–10% B in 5–35 min, 10% B in 35–45 min, and initialized to 50% B in 45–50 min. Protonated ions of nucleosides were scanned in a positive polarity mode over an $m/z$ range of 110–700.

For RNA fragment analysis, the isolated tRNAs (2 pmol) were digested with RNase T$_1$ (10 U; Thermo Fisher Scientific) in 25 mM ammonium acetate (pH 5.3) or RNase A (10 ng; Thermo Fisher Scientific) in 25 mM ammonium acetate (pH 7.3) at 37 °C for 60 min. The digests were mixed with one-tenth volume of 0.1 M triethylamine acetate (pH 7.0), and subjected to capillary LC/nano ESI-MS on an LTQ Orbitrap mass spectrometer (Thermo Fisher Scientific) equipped with a splitless nanoflow high-performance LC (nano-HPLC) system (DiNa; KYA Technologies) using a nano-LC trap column and a capillary column (HiQ Sil C18W-3, 0.1 × 100 mm; KYA Technologies)[57,94]. The digested tRNA fragments were separated at a flow rate of 300 nL/min by capillary LC in a solvent system consisting of 0.4 M 1,1,1,3,3,3-hexafluoro-2-propanol (HFIP) (pH 7.0) (solvent A) and 0.4 M HFIP (pH 7.0) in 50% methanol (solvent B) using a linear gradient (2–100% B in 40 min)[95]. The eluent was ionized by an ESI source in negative polarity mode and scanned over an $m/z$ range of 600–2000. The CID was performed by data-dependent scan with normalized collision energy 40. The product ions were assigned as described[96].

Qual Browser software bundled in Xcalubur (Thermo Fisher Scientific) was used for data analysis.

**Measuring frequency of mt-tRNA modification**. In negative mode of ESI, detection efficiencies of the RNA fragments having the same sequence but different modification status do not differ largely, because ESI ionization relies mainly on numbers of phosphate groups, not on type of base modifications[28,97]. Hence, we are able to relatively quantify frequency of tRNA modification at each site from the extracted-ion chromatogram (XICs) intensities between the modified and unmodified RNA fragments. We analyzed both modified and unmodified fragments for each site and calculated the modification frequencies (Supplementary Data 1 and 2). Due to technical difficulties, we could not calculate frequencies of all the modification detected in this study. We here show modification frequencies for 51 sites in 15 species of mt-tRNAs (Supplementary Data 1 and 2).

**CE-Ψ-MS**. Cyanoethylation of tRNA was performed essentially as described[8,56] with slight modification. The isolated tRNA (5–10 pmol) dissolved in ≤4 μL water was mixed with 30 μL of 41% (v/v) ethanol/1.1 M trimethylammonium acetate (pH 8.6) and 4 μL of acrylonitrile (Fujifilm Wako Pure Chemical Corporation), and then incubated at 70 °C for 2 h. The cyanoethylated tRNA was collected by ethanol precipitation and dissolved in water. The solution was then subjected to RNase digestion and analyzed by RNA mass spectrometry as described above.

**CMC-PE**. Detection of Ψ sites by CMC derivatization and primer extension was performed essentially as described[60,98,99] with slight modification. HeLa total RNA (10 μg) was dissolved in 30 μL of CMC solution [0.17 M *N*-cyclohexyl-*N′*-(2-morpholinoethyl)carbodiimide methyl-*p*-toluenesulfonate (CMC), 7 M urea, 50 mM Bicine, and 4 mM EDTA] and incubated at 37 °C for 20 min. The reaction was quenched by addition of 100 μL stop solution [0.3 M NaOAc (pH 5.6) and 0.1 M EDTA], followed by addition of 700 μL ethanol to precipitate the RNA. The RNA pellet was rinsed twice with 70% EtOH and dried. The CMC-derivatized RNA was dissolved in 40 μL of 50 mM Na$_2$CO$_3$ (pH 10.4), incubated at 37 °C for 4 h (alkaline treatment), ethanol precipitated, and dissolved in 10 μL water. The treated RNA (1 μg) was mixed with the 5′ $^{32}$P-labeled primer (0.1 pmol) in a 5 μL solution containing 10 mM Tris-HCl (pH 8.0) and 1 mM EDTA, incubated at 80 °C for 2 min, and cooled to room temperature. Subsequently, reverse transcription was started by adding 0.75 μL water, 2 μL of 5× FS buffer (Invitrogen), 0.25 μL of 1.5 mM dNTP mix, 1.5 μL of 25 mM MgCl$_2$, and 0.5 μL of SuperScript III (Invitrogen), and the mixture was incubated at 55 °C for 1 h. As a control, the same reaction was performed with total RNA untreated with CMC. The RNA sequencing ladder was generated by reverse transcription under the same condition in the presence of 37.5 μM of ddTTP or ddATP. To stop the reaction and degrade the template RNA, 0.5 μL of 4 M NaOH was added to the reaction mixture, followed by boiling at 95 °C for 5 min, and then the reaction was neutralized by addition of 9 μL of 1 M Tris-HCl (pH 5.0). cDNAs were resolved on 12% PAGE with 7 M urea. The radiolabeled bands were visualized by fluoroimager (FLA-7000; Fujifilm). Primer sequences are listed in Supplementary Data 5.

**tRNA-Ψ-seq**. tRNA-Ψ-seq was basically performed essentially as described[100–102] with several modifications. Total RNA was extracted from HEK293T cells by acid guanidinium thiocyanate-phenol-chloroform extraction (AGPC) method[103]. The tRNA fraction was separated by 10% PAGE with 7 M urea, excised from the gel and eluted from the gel pieces, subjected to deacylation with 0.1 M CHES-NaOH buffer (pH 9.0) at 37 °C for 30 min, and ethanol precipitated. The deacylated tRNA was then demethylated by *E. coli* AlkB at 37 °C for 3 h in a reaction mixture consisting of 45 mM Tris-HCl (pH 8.0), 67 μM (NH$_4$)$_2$Fe(SO$_4$)$_2$·6H$_2$O, 0.9 mM 2-ketoglutarate, 1.8 mM L-ascorbic acid, 10 ng/μL tRNA, 1.25 μM *E. coli* wild-type AlkB, and 0.125 μM *E. coli* AlkB L118V/D135S mutant. The demethylated RNA was extracted by phenol–chloroform–isoamylalcohol (PCI) and chloroform, followed with ethanol precipitation.

The demethylated RNA (5 μg) was dissolved in 15 μL of CMC-BEU buffer [0.34 M CMC, 7 M urea, 4 mM EDTA, 50 mM bicine (pH 7.88)] and incubated at 37 °C for 20 min. For the CMC-minus control sample, the same reaction was

performed without CMC. The reaction was quenched by addition of 100 μL stop solution [0.3 M NaOAc (pH 5.2), 0.1 M EDTA]. After ethanol precipitation, the tRNA pellet was dissolved in 40 μL of 50 mM sodium carbonate buffer (pH 10.4) and incubated at 37 °C for 4 h to remove CMC from U or G bases. The CMC-treated and untreated tRNAs were 3′ dephosphoryled at 37 °C for 1 h in a mixture containing 100 mM Tris-Hcl (pH 6.8), 10 mM MgCl₂, 2 mM DTT, and 0.2 U/μL T4 polynucleotide kinase (NEB), and then subjected to PCI and chloroform extractions and ethanol precipitation. The prepared RNA (200 ng) was ligated with 3′ RNA linker at 25 °C for 16 h in a mixture containing 1 × T4 RNA Ligase Reaction Buffer (NEB), 1 μM Universal miRNA cloning linker (NEB), 15% PEG8000, 2 U/μL SUPERase•In (Invitrogen), and 10 U/μL T4 RNA ligase 2, truncated KQ (NEB), followed by PCI and chloroform extractions and ethanol precipitation. The ligated product was resolved and separated from the excess 3′ adaptor by running 10% PAGE with 7 M urea, excised from the gel, eluted from the gel pieces, and ethanol precipitated.

Next, half of the adaptor-ligated tRNA was dissolved in 11.2 μL water and 0.8 μL RT primer (3.75 μM) (Supplementary Data 6) was added, denatured at 80 °C for 5 min, and placed at room temperature for several minutes. Then, 8 μL TGIRT Reaction Buffer [Tris-HCl pH 7.5 (20 mM, final), 5 mM MgCl₂ (20 mM, final), NaCl (450 mM, final), dNTPs (1 mM, final), DTT (5 mM, final), SUPERase•In (1 U/μL, final), and TGIRT-III (InGex, 5%, final)] were added to the denatured tRNA-primer mix, and reverse transcription was performed at 57 °C for 2 h. The reaction was quenched, and the template tRNA was degraded by addition of 1.1 μL of 2 M NaOH and incubation at 95 °C for 5 min, followed by ethanol precipitation. The first-strand cDNA was resolved by running 6% PAGE with 7 M urea, excised from the gel, eluted from the gel pieces, and ethanol precipitated.

Half of the cDNA was circularized at 60 °C for 18 h in a mixture containing 1× CircLigase Reaction buffer, 2.5 mM MnCl₂, 50 μM ATP, and 5 U/μL CircLigase II (Lucigen), followed by heat inactivation at 80 °C for 10 min. The circularized cDNA was extracted by PCI and chloroform extractions, and ethanol precipitated. One-fourth of the circularized cDNA was PCR-amplified with 15 cycles in a mixture containing 1 × Phusion GC Buffer, 0.2 mM dNTPs, 0.5 μM forward and reverse PCR primer (Supplementary Data 6), 3% DMSO, and 20 U/mL phusion polymerase (NEB). The PCR product was then collected with AMPure XP (Beckman Coulter), resolved by 6% native PAGE, excised from the gel, eluted from the gel pieces, and purified by AMPure XP. The cDNA libraries were sequenced in 100-bp paired-end mode on an Illumina HiSeq4000 platform.

**Data analysis of tRNA-Ψ-seq.** Raw reads from Illumina sequencing were subjected to adaptor trimming and filtering of low-quality reads by fastp 0.20.0 (https://github.com/OpenGene/fastp). The minimum length for reads after trimming was set to 15 nucleotides, and the quality threshold was set to Q15. The trimmed reads were mapped to the human mitochondrial tRNA sequence (obtained from NCBI and manually curated) with 10 Ns on both sides using Bowtie2 version.2.3.3.1(http://bowtie-bio.sourceforge.net/bowtie2/index.shtml). Properly paired reads that overlapped with the CCA terminal sequence of tRNA for at least one nucleotide and did not overlap with the 3′ trailer sequence were used for the Ψ score calculation. A summary of trimming and mapping is provided in Supplementary Data 7. Based on the 5′ end position and mutations in the mapped reads, the Ψ score at the reference position $x$ was obtained by the following Eq. (1).

$$\Psi \text{ score} = \left( \left( \frac{rm(x)}{rm(x+1)} - \frac{rp(x)}{rp(x+1)} \right) - \left( \frac{rm(x+1)}{rm(x+2)} - \frac{rp(x+1)}{rp(x+2)} \right) \right) \times 100. \tag{1}$$

Function $rm(x)$ returns the number of reads piled up at position $x$ in the CMC-minus sample, and function $rp(x)$ returns the number in the CMC-plus sample. When the leftmost or penultimate nucleotide is a mismatch against the reference, the 5′ end of the read is shifted to the 3′ side of the mismatch. The final Ψ score is the average of the two replicates. The calculations were carried out by a Python script (http://python.org), utilizing the libraries Pysam (https://github.com/pysam-developers/pysam) and Biopython[104]. All the custom scripts used in this analysis are available upon request.

**Post-labeling method to determine the tRNA modifications.** The post-labeling method was basically performed as instructed in a procedure[105]. In brief, purified tRNA$^{Glu}$ was partially hydrolyzed by heat treatment. The 5′ terminus of each fragment in the hydrolysate was $^{32}$P-phosphorylated with T4 polynucleotide kinase (Toyobo). The radiolabeled fragments were resolved on 12% PAGE with 7 M urea and visualized on a fluoroimager (FLA-7000, Fujifilm), and each band was excised from the gel. The fragment was eluted from the gel piece and digested with nuclease P1. The $^{32}$P-labeled 5′-terminal nucleotide of each fragment was analyzed by 2D-TLC with cellulose plates using the solvent system consisting of isobutyric acid/concentrated ammonia/H₂O (66:1:33 v/v/v) in the first dimension and 2-propanol/HCl/H₂O (70:15:15 v/v/v) in the second dimension. The radiolabeled spots on the TLC plates were visualized on a fluoroimager (FLA-7000, Fujifilm). The corresponding modified nucleotide was determined by referring to a 2D-TLC map of modified nucleotides[106].

**Construction of *QTRT1* or *QTRT2* KO cell lines.** *QTRT1* or *QTRT2* KO HEK293T cells were generated using the CRISPR/Cas9 system[94,107]. In short, sense and antisense oligonucleotides corresponding to the single guide RNA (sgRNA) (Supplementary Data 6) were cloned into pX330 vector (Addgene #42230)[108]. HEK293T cells seeded in 24-well plates were transfected using FuGENE HD (Promega) with 300 ng pX330 containing the designed sgRNA sequence, 100 ng of pEGFP-N1 (Clontech) to monitor transfection efficiency by EGFP fluorescence, and 100 ng of a modified pLL3.7 containing a puromycin-resistance gene. One day after transfection, cells were seeded at low density, and transfectants were selected with 1 μg/mL puromycin. Eight days after transfection, several colonies were picked and grown for several days. The genomic region targeted by the sgRNA in each clone was PCR-amplified and sequenced; primers are listed in Supplementary Data 6. WT and the KO HEK293T cell lines were cultured at 37 °C in 5% CO₂ in DMEM (D5796, Sigma-Aldrich) supplemented with 5% fetal bovine serum (Gibco), 1% penicillin–streptomycin (Fujifilm Wako Pure Chemical Corporation), and 140 nM queuine.

**Mitoribosome profiling**
*Library preparation.* The cell extracts were prepared with lysis buffer[94,109] supplemented with 100 μg/mL chloramphenicol, treated with 15 U of TURBO DNase (Thermo Fisher Scientific), and clarified by centrifugation at 20,000 × g, 4 °C for 10 min. The RNA content in the lysate was measured with the Qubit RNA BR Assay Kit (Thermo Fisher Scientific). The lysate containing 10 μg of RNA was digested with 20 U of RNase I (Lucigen). The ribosomes were pelleted by sucrose cushion with ultracentrifugation. After RNA extraction with TRIzol (Thermo Fisher Scientific) and Direct-zol RNA microprep kit (Zymo research), the protected RNA fragments ranging from 17 to 34 nt were gel-excised. Following, dephosphorylation, linker ligation, reverse transcription, and circularization[94,109], the libraries were PCR-amplified and sequenced on an Illumina HiSeq4000 platform (single-end 50 bp read). See Supplemental Data 6 for the details for DNA/RNA oligonucleotides used in the library preparation.

*Data analysis.* After trimming the linker sequences, all reads were first aligned to human non-coding RNAs (rRNAs, tRNAs, mt-rRNAs, mt-tRNAs, snRNAs, snoRNAs, and miRNAs) by STAR[110]. Then, the un-aligned reads were mapped to the custom-made mitochondrial transcript sequences[2,111] by STAR. On the mitochondrial footprints, the distances from 5′ end to the ribosomal A site were estimated as follows: 14 for 23- to 31-nt long reads and 15 for 32 to 34-nt long reads. Then, the reads on given A-site codon were assigned, using Riboseq tools (https://github.com/ingolia-lab/RiboSeq).

For calculation of codon occupancies, the reads at each codon were normalized with averaged reads per codon in the transcript. Codon positions with codon occupancy lower than 5% in each transcript in any two or more samples among WT replication 1, WT replication 2, *QTRT2* KO replication 1, and *QTRT2* KO replication 2 were removed from analysis. Start and stop codons were also omitted from the analysis. All the custom scripts used in this study are available upon request.

**Protein aggregation assay.** The reporter plasmids encoding firefly luciferase fused with enhanced green fluorescent protein (Fluc-EGFP) and its structurally desta-bilized mutants, FlucSM-EGFP (R188Q mutant) and FlucDM-EGFP (R188Q/R261Q mutant), were obtained from Addgene (#90170–#90172, respectively)[72]. Wild type and *QTRT2* KO HEK293T cells (3.5 × 10⁵ cells/well) were cultured in six-well plates (IWAKI) for 24 h, followed by transfection with 3 μg of each reporter plasmid using FuGENE HD (Promega). After 48 h, ten fluorescence images per well were acquired using a fluorescence microscope (AF6500 System, Leica Microsystems) equipped with an objective lens (HCX PL FLUOTAR 10×/0.30 DRY; Leica Microsystems) in the setting as follows: Chanel, GFP; Exposure time, 100 ms; Gain, 1; Intensity, 5. To count the number of foci which represent reporter protein aggregates, the images were processed by ImageJ software. The macrocode was set as follows: run("Subtract Background…", "rolling=15"); run ("8-bit"); setAutoThreshold("Default dark"); setThreshold(33, 255); setOption ("BlackBackground", true); run("Convert to Mask"); run("Analyze Particles…", "include summarize"). To estimate transfection efficiency, the area of EGFP fluorescence was obtained by the same macro code as above, except setThreshold (33, 255) was replaced by setThreshold(4, 255). Aggregation rate was calculated as the number of aggregates divided by the area of EGFP fluorescence. Statistical analyses were performed with R.

**Reporting summary.** Further information on research design is available in the Nature Research Reporting Summary linked to this article.

## Data availability

The source data underlying Figs. 3c, 6d and Supplementary Fig. 4 are provided as a Source Data file. Data for tRNA-Ψ-seq and ribosome profiling generated in this study have been deposited in NCBI SRA (PRJNA638467) and GEO (GSE150439), respectively. The tRNA sequences including modifications have been deposited in DDBJ with the accession codes LC530712, LC530713, LC530714, LC530715, LC530716, LC530717, LC530718, LC530719, LC530720, LC530721, LC530722, LC530723, LC530724,

LC530725, and LC530726 for mt-tRNAs for Ala, Cys, Glu, Phe, Gly, His, Met, Asn, Pro, Gln, Arg, Thr, Val, Trp, and Tyr, respectively. tRNA sequences are also accessible at our web site: http://rna.chem.t.u-tokyo.ac.jp/trnadata/hsmttrna/mttrnas.html. All data supporting the findings in this study are available from the corresponding author upon reasonable request. Source data are provided with this paper.

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

## Acknowledgements

We are grateful to the members of the Suzuki laboratory, in particular, K. Miyauchi and A. Nagao, for many insightful suggestions and discussions, and S. Kitagawa and L. Kawarada for providing materials. We are also grateful to S. Nishimura (University of Tsukuba) for providing queuine. Special thanks are due to Tao Pan for providing useful information. This work was supported by Grant-in-Aid for Scientific Research from Ministry of Education, Culture, Sports, Science and Technology (MEXT) and the Japan Society for the Promotion of Science (JSPS) (to Takeo Suzuki and Tsutomu Suzuki). This work used the Vincent J. Coates Genomics Sequencing Laboratory at the University of California Berkeley, supported by NIH S10 OD018174 Instrumentation Grant. Computations were supported by Bioinformatics Analysis Environment Service on RIKEN Cloud at RIKEN Advanced Center for Computing and Communications and partially performed on the NIG supercomputer at the ROIS National Institute of Genetics.

## Author contributions

Takeo Suzuki, Y.Y., I.M., D.M., X.Z., K.A., H.L., and Y.K. performed biochemical, genetic, and cellular analyses. Y.S. performed mass spec analyses. H.S. and M.M. performed ribosome profiling assisted by S.O. and S.I. I.K. and Y.I. performed tRNA-Ψ-seq. All authors discussed the results. Takeo Suzuki and Tsutomu Suzuki wrote this paper. Tsutomu Suzuki supervised all the work.

## Competing interests

The authors declare no competing interests.
