## [Peer Review FIle · Nature Communications]

Reviewers' comments:

Reviewer #1 (Remarks to the Author):

tRNAs are central to the decoding process and the post-transcriptional modifications of tRNA are critical for tRNA function including stability, folding and decoding. It is known that disorders of mitochondrial translation may lead to the mitochondria dysfunction and some point mutations in mt-tRNA have been reported to be associated with mitochondrial diseases. But the molecular pathogenesis of human mitochondrial diseases caused by aberrant tRNA modifications has lagged behind due to the lack of the complete landscape of modifications in mt-tRNA and the list of responsible modifying enzymes. In this manuscript, Suzuki and his colleagues identified the majority of modifications in all 22 human mt-tRNA using mass spectrometric analyses. Further, a comprehensive analysis including CE- Ψ -MS, CMC-PE and tRNA- Ψ -seq were employed to detect Ψ (a mass-silent modification) in mt-tRNA. In addition, the modifying enzymes responsible for these mt-tRNA are also listed. Thus, the collective results represent a very comprehensive resource that will be a key reference for the field. The study will certainly benefit future molecular mechanism studies of human mitochondrial diseases caused by mt-tRNA modifications. The study is also very well designed, performed and written. Hence, I strongly recommend the publication of this paper at Nature communications. I only have a few minor points as listed.

1. In this study, tRNA- Ψ -seq was used to detect Ψ sites in mt-tRNA. Recently a similar approach named DM- Ψ -seq is also developed and identified Ψ sites in cytoplasmic and mitochondrial tRNA (Song et al., 2020, Nat Chem Biol 16, 160-169). It would be nice to also compare the Ψ sites (since it is a mass-silent modification) identified in this study with the reference.
2. The authors have listed RNase T1- or A-digested fragments of 22 human mt-tRNAs in supplementary table 1; all the original fragment analysis maps (as Figure 2C) are also desired to be included, which can be used to evaluate the stoichiometry of modified nucleosides.
3. 2D-TLC cannot discriminate Ψ in different positions of mt-tRNA^{Glu} and hence the label in Supplementary 4 is not accurate.

Reviewer #2 (Remarks to the Author):

In this study, Suzuki et al, performed a comprehensive analysis of post-transcriptional modifications of all human mt-tRNAs. In a previous analysis, the authors mapped 15 types of modified bases at 118 positions in all species of bovine mt-tRNAs. Here, the authors isolated 22 species of human mt-tRNAs and analyzed the primary sequence and post-transcriptional modifications for each tRNA using mass

spectrometry in combination with biochemical analyses to map pseudouridines. In total, the authors found 16 kinds of RNA modifications at 136 positions in 22 species of human mt-tRNAs. The results provide a complete list of modifications in human mt-tRNA. This is an interesting study and should facilitate investigations of their physiological roles in the future.

1. The authors isolated 22 species of human mt-tRNAs from human placenta, but only several species of mt-tRNA from HeLa cells. Why not isolate the all 22 species of human mt-tRNA from HeLa cells? I suggest the authors to compare the modifications in mt-tRNA between human placenta and HeLa cells.

2. How about the frequency (or stoichiometry) of specific modifications in given sites of human mt-tRNA? I suggest the authors to provide this information for each identified modification in mt-tRNA.

3. How to confirm the purity of the isolated individual mt-tRNA? The lengths of mt-tRNA are similar and they also show similar gel-shift on the gel. Single band in gel may not indicate the single mt-tRNA. The impure mt-tRNA may affect the determination of the modifications in mt-tRNA.

4. There are some modifications that have the same molecular weights, such as m5C and m3C, m1G and m2G, m1A and m6A. How to confirm these modifications? The authors need to provide detailed experimental evidence.

5. The MS fragment ions of 15 species of mt-tRNA were listed in Table S1. But in this study the authors measured 22 species of mt-tRNA. The authors need to supply the rest part of mt-tRNA of the MS fragment ions.

6. The authors need to provide the sequence and Western blotting data for supporting the knockout experiments.

7. In Figure 3b, the Ψ s at positions 66 and 67 in human mt-tRNA^{Pro} were identified. If these two sites are both Ψ s, why there are two possibilities, i.e. the existence of c6 and y6 ions?

8. Figure 4 and Figure S5 are somehow similar. Figure 4 includes 14 species of mt-tRNA and Figure S5 includes 22. The authors may only provide the full 22 species of mt-tRNA in main text.

Reviewer #3 (Remarks to the Author):

In this manuscript by Suzuki et al., the authors map the chemical structures of the remaining uncharacterized tRNAs encoded by the human mitochondrial genome. Using previously-developed tRNA purification and mass spectrometry techniques, the authors identify the repertoire of modifications in human mitochondrial tRNAs. The data is technically sound and the authors provide independent assays to validate a subset of modifications. Moreover, the authors provide evidence that the QTRT1 and QTRT2 proteins are required for the queuosine modification in certain human mitochondrial tRNAs. The

manuscript is clearly written and have provided extensive Supplemental Data to support their findings.

Unfortunately, there is a lack of novelty in this manuscript since no new types of modifications were found. The approach and results are similar to the conclusions published by the group using bovine mitochondrial tRNAs in their 2014 paper in *Nucleic Acids Research*. Thus, many of the modifications in mitochondrial tRNA could have been predicted and this manuscript serves as validation rather than discovery. Moreover, there is little functional characterization of the modifications and their biological roles. Table 1 is just a list of codons and the cognate tRNA anticodons while Table 2 is a list of the known and confirmed tRNA modifications and their cognate enzymes. While the results represent a complete inventory of human mitochondrial tRNA modifications, no new insight is gained that changes how we think about mitochondrial tRNA modifications or mitochondrial tRNA biology.

To elevate this manuscript beyond just simply mapping tRNA modifications, the authors should characterize the consequences of lacking a mitochondrial tRNA modification. For example, the Authors have made a knockout of QTRT1 or QTRT2 but there is no follow-up on the function of queuosine modification in mitochondrial tRNAs. Biological insight could also be gained by monitoring the function of the numerous pseudouridine modifications in mitochondrial tRNAs. Currently, the manuscript describes a list of modifications without providing any biological relevance or significance of the modifications.

Minor points:

- Figure 1 shows a denaturing gel of isolated tRNAs. Since the RNAs are imaged using a general nucleic acid stain, there is no follow-up on the relative purity of the isolated tRNA fractions. The Authors could perform Northern blotting on an aliquot of the isolated tRNAs to determine whether there are contaminating tRNAs in the fractions. This could be important since any contaminating tRNAs could contribute significantly to the modification signal detected by mass spectrometry.
- As mentioned above, there is no follow-up on the QTRT1 or QTRT2-knockout (KO) cell lines. The addition of molecular or biological effects caused by lack of Q modification would have greatly strengthened the manuscript and provided insight into the function of the Q modification in mitochondrial tRNAs.

First of all, we appreciate all reviewers for spending precious time to review our manuscript, and giving us a number of productive comments to improve it. Revised descriptions in the main text are marked.

Response to Reviewer #1's comments

Reviewer #1 (Remarks to the Author):

tRNAs are central to the decoding process and the post-transcriptional modifications of tRNA are critical for tRNA function including stability, folding and decoding. It is known that disorders of mitochondrial translation may lead to the mitochondria dysfunction and some point mutations in mt-tRNA have been reported to be associated with mitochondrial diseases. But the molecular pathogenesis of human mitochondrial diseases caused by aberrant tRNA modifications has lagged behind due to the lack of the complete landscape of modifications in mt-tRNA and the list of responsible modifying enzymes. In this manuscript, Suzuki and his colleagues identified the majority of modifications in all 22 human mt-tRNA using mass spectrometric analyses. Further, a comprehensive analysis including CE- Ψ -MS, CMC-PE and tRNA- Ψ -seq were employed to detect Ψ (a mass-silent modification) in mt-tRNA. In addition, the modifying enzymes responsible for these mt-tRNA are also listed. Thus, the collective results represent a very comprehensive resource that will be a key reference for the field. The study will certainly benefit future molecular mechanism studies of human mitochondrial diseases caused by mt-tRNA modifications. The study is also very well designed, performed and written. Hence, I strongly recommend the publication of this paper at Nature communications. I only have a few minor points as listed.

We really appreciate the positive comments that encourage us very much.

1. In this study, tRNA- Ψ -seq was used to detect Ψ sites in mt-tRNA. Recently a similar approach named DM- Ψ -seq is also developed and identified Ψ sites in cytoplasmic and mitochondrial tRNA (Song et al., 2020, Nat Chem Biol 16, 160-169). It would be nice to also compare the psiU sites (since it is a mass-silent modification) identified in this study with the reference.

We also notice the DM- Ψ -seq paper by Yi's group. It is interesting for us to compare the data. But, unfortunately, there is no data available for Ψ sites in mitochondrial tRNAs in this paper.

2. The authors have listed RNase T1- or A-digested fragments of 22 human mt-tRNAs in

supplementary table 1; all the original fragment analysis maps (as Figure 2C) are also desired to be included, which can be used to evaluate the stoichiometry of modified nucleosides.

In this manuscript, we showed mass spec data for 15 species of human mt-RNAs, not 22 species, because rest of them were already reported. As requested, we here show all the original fragment analysis maps for the rest of 14 mt-tRNA species (Supplementary Figure 1ab). In addition, we measured the frequency (stoichiometry) for each tRNA modification as much as possible (Figures 2e and 5a-f and Supplementary Tables 1 and 2). Please understand that we cannot analyze all the modifications contained in each tRNA because of technical difficulties.

3. 2D-TLC cannot discriminate Ψ in different positions of mt-tRNA^{Glu} and hence the label in Supplementary 4 is not accurate.

This is a post-labeling method to detect tRNA modifications at specific positions.

Response to Reviewer #2's comments

Reviewer #2 (Remarks to the Author):

In this study, Suzuki et al, performed a comprehensive analysis of post-transcriptional modifications of all human mt-tRNAs. In a previous analysis, the authors mapped 15 types of modified bases at 118 positions in all species of bovine mt-tRNAs. Here, the authors isolated 22 species of human mt-tRNAs and analyzed the primary sequence and post-transcriptional modifications for each tRNA using mass spectrometry in combination with biochemical analyses to map pseudouridines. In total, the authors found 16 kinds of RNA modifications at 136 positions in 22 species of human mt-tRNAs. The results provide a complete list of modifications in human mt-tRNA. This is an interesting study and should facilitate investigations of their physiological roles in the future.

We appreciate these positive words.

1. The authors isolated 22 species of human mt-tRNAs from human placenta, but only several species of mt-tRNA from HeLa cells. Why not isolate the all 22 species of human mt-tRNA from HeLa cells? I suggest the authors to compare the modifications in mt-tRNA between human placenta and HeLa cells.

This is the first resource paper aiming to map all the modifications contained in each mt-

tRNA. So, comparing each modification between tissues or cells is out of scope in this paper, and it should be done in future studies. As you imagine, it will be a very difficult and time-consuming work to isolate all the mt-tRNA species from HeLa cells, because steady-state level of mt-tRNA is quite low. At this moment, we have a dataset of mt-tRNA^{Gln} isolated from both human placenta and HeLa cells, and compared tRNA modification status between them (Figure A). The same set of species and position of tRNA modifications are detected in both samples, but frequency of each modification is slightly different.

2. How about the frequency (or stoichiometry) of specific modifications in given sites of human mt-tRNA? I suggest the authors to provide this information for each identified modification in mt-tRNA.

This is the similar request of reviewer #1. We measured the frequency of modification in given sites as much as possible (Figure 5 and Supplementary Tables 1 and 2). Please

understand that we cannot analyze all the modifications contained in each tRNA because of technical difficulties.

3. How to confirm the purity of the isolated individual mt-tRNA? The lengths of mt-tRNA are similar and they also show similar gel-shift on the gel. Single band in gel may not indicate the single mt-tRNA. The impure mt-tRNA may affect the determination of the modifications in mt-tRNA.

Yes. Single band does not ensure purity of tRNA. As shown in our mass spec data (Figure 2cd, Supplementary Figure 1 and Table 1), most of RNA fragments including modifications were unequivocally assigned to each individual tRNA. The quality of tRNA isolation is assured. We usually confirm the purity of tRNA by mass spec. The tRNA isolation method we used here has been established very well, and applied to many projects from my laboratory and collaborators.

4. There are some modifications that have the same molecular weights, such as m⁵C and m³C, m¹G and m²G, m¹A and m⁶A. How to confirm these modifications? The authors need to provide detailed experimental evidence.

We first profile total nucleosides to know species of modifications contained in each tRNA. Then, we assign each modification to a specific position of tRNA by the fragment analyses using RNase T₁ and A. Most of modifications are assigned by this procedure. For m⁵C and m³C, there is no tRNA having both modifications by the nucleoside analyses. For m¹G and m²G, we can assign them by different sensitivity to RNase T₁. m²G is a good substrate, whereas m¹G is resistant to the cleavage. If m¹G is digested by RNase T₁, its 3' terminus always shows 2',3'-cyclic phosphate. From this feature, we are able to distinguish them. For m¹A and m⁶A, m¹A is present at positions 9, 16 and 58, and their responsible enzymes have been determined. However, there is no report and evidence of m⁶A in human mt-tRNAs so far. As written in the main text, although we detected small amount of m⁶A in the nucleoside analysis, it is likely that m⁶A is artificially produced by the Dimroth reaction during the sample preparation.

5. The MS fragment ions of 15 species of mt-tRNA were listed in Table S1. But in this study the authors measured 22 species of mt-tRNA. The authors need to supply the rest part of mt-tRNA of the MS fragment ions.

In this manuscript, we showed mass spec data for 15 species of human mt-RNAs, not 22 species, because rest of them were already reported.

6. The authors need to provide the sequence and Western blotting data for supporting the knockout experiments.

We provide electropherogram of Sanger sequence of the target region of *QTRT1* KO and *QTRT2* KO (Figure 6A). We don't have any reliable antibodies to detect endogenous QTRT1 and QTRT2 so far.

7. In Figure 3b, the Ψ s at positions 66 and 67 in human mt-tRNA^{Pro} were identified. If these two sites are both Ψ s, why there are two possibilities, i.e. the existence of c6 and y6 ions?

Because this is a CID analysis of the RNA fragment containing one cyanoethylation, we detected two different fragments, one has ce¹ Ψ 66, the other has ce¹ Ψ 67 (Figure 3b), indicating that Ψ is partially introduced at these two sites with low frequency. In fact, we detected the same fragment having two cyanoethyl groups, but its abundance is too low to obtain any CID data for it.

8. Figure 4 and Figure S5 are somehow similar. Figure 4 includes 14 species of mt-tRNA and Figure S5 includes 22. The authors may only provide the full 22 species of mt-tRNA in main text.

Because rest of tRNAs have been already reported, we would like to show 14 species in the main figure.

Response to Reviewer #3's comments

Reviewer #3 (Remarks to the Author):

In this manuscript by Suzuki et al., the authors map the chemical structures of the remaining uncharacterized tRNAs encoded by the human mitochondrial genome. Using previously-developed tRNA purification and mass spectrometry techniques, the authors identify the repertoire of modifications in human mitochondrial tRNAs. The data is technically sound and the authors provide independent assays to validate a subset of modifications. Moreover, the authors provide evidence that the QTRT1 and QTRT2 proteins are required for the queuosine modification in certain human mitochondrial tRNAs. The manuscript is clearly written and have provided extensive Supplemental Data to support their findings.

Unfortunately, there is a lack of novelty in this manuscript since no new types of

modifications were found. The approach and results are similar to the conclusions published by the group using bovine mitochondrial tRNAs in their 2014 paper in *Nucleic Acids Research*. Thus, many of the modifications in mitochondrial tRNA could have been predicted and this manuscript serves as validation rather than discovery.

I understand this reviewer's impression and concerns. Let me explain the novelty and value of this manuscript. Even though we had a complete list of modifications in bovine mt-tRNAs, it is impossible to predict exact positions of modifications in human mt-tRNAs, because mt-tRNA sequences are quite different between these mammals. They share only 81% sequence identity. To study molecular pathogenesis of mitochondrial diseases due to lack of tRNA modification, it is necessary to determine exact species and positions of all the modifications in each human mt-tRNA.

After careful analysis of our mass spec data, we have detected small amount of 5-carboxymethylaminomethyluridine (cmnm⁵U, 0.6%) in mt-tRNA^{Trp} (Fig. 5b), and carboxymethylaminomethyl-2-thiouridine (cmnm⁵s²U, 1.3%) in mt-tRNA^{Gln} (Fig. 5c). They are minor but additional modifications that have never been detected in bovine mt-tRNAs. Now we have 18 species of RNA modifications in human mt-tRNAs. We previously reported that these modifications were observed in mt-tRNAs under taurine depletion (Asano et al., 2018). But, in this study, cmnm⁵(s²)U are naturally present in mt-tRNAs in HeLa cell cultured under normal growth condition as well as in human placenta.

Moreover, there is little functional characterization of the modifications and their biological roles. Table 1 is just a list of codons and the cognate tRNA anticodons while Table 2 is a list of the known and confirmed tRNA modifications and their cognate enzymes. While the results represent a complete inventory of human mitochondrial tRNA modifications, no new insight is gained that changes how we think about mitochondrial tRNA modifications or mitochondrial tRNA biology.

To elevate this manuscript beyond just simply mapping tRNA modifications, the authors should characterize the consequences of lacking a mitochondrial tRNA modification. For example, the Authors have made a knockout of QTRT1 or QTRT2 but there is no follow-up on the function of queuosine modification in mitochondrial tRNAs. Biological insight could also be gained by monitoring the function of the numerous pseudouridine modifications in mitochondrial tRNAs. Currently, the manuscript describes a list of modifications without providing any biological relevance or significance of the modifications.

This manuscript is a resource paper describing a complete inventory of human mitochondrial tRNA modifications, not a functional analysis paper. Biological relevance of each

modification should be studied in another paper in the future. A complete list of human mt-tRNA modifications based on reliable datasets will be valuable and necessary to study human mitochondrial physiology and pathology. We show Table 1 for the first time by determining all the anticodons including modifications. Although this is the same table as we reported for bovine mt-tRNA previously, this is the first report of the codon-anticodon pairing rule in human mitochondria based on the concrete evidence. In addition to the complete list of tRNA modifications, we are able to list their responsible enzymes/genes including candidate genes (Table 2). In addition, we have revealed that 23 pathogenic point mutations coincide with the tRNA modification sites (Supplementary Table 5), implying that loss of tRNA modification due to such pathogenic mutation can be a primary cause of these mitochondrial diseases.

Although this is a resource paper, we added some follow-up data on the function of Q34 in mt-tRNAs, as requested by this reviewer. We have carried out mitoribosome profiling of *QTRT2* KO versus wild type HEK293T cells, and calculated codon occupancy change upon *QTRT2* KO. As shown in Figure 6c, the codon occupancy at UAU codon increased significantly upon *QTRT2* KO, indicating that Q34 in mt-tRNA^{Tyr} plays a role in efficient decoding of UAU codon in mitochondria.

In addition, we found a protein folding problem due to lack of codon optimality in *QTRT2* KO cells (Figure 6d). This is an apparent phenotype in the absence of Q34 modification in both cytoplasmic and mitochondrial tRNAs. Of course, it is difficult to estimate the effect of Q34 deficiency in mt-tRNAs on the cytoplasmic protein homeostasis. Our group previously reported that $\tau^5\text{U}34$ deficiency in mt-tRNAs induced protein aggregation in cytoplasm (Fakruddin et al., 2018). This fact naturally speculates that Q34 deficiency in mt-tRNAs could affect cytoplasmic protein homeostasis to some extent.

Minor points:

- Figure 1 shows a denaturing gel of isolated tRNAs. Since the RNAs are imaged using a general nucleic acid stain, there is no follow-up on the relative purity of the isolated tRNA fractions. The Authors could perform Northern blotting on an aliquot of the isolated tRNAs to determine whether there are contaminating tRNAs in the fractions. This could be important since any contaminating tRNAs could contribute significantly to the modification signal detected by mass spectrometry.

As shown in our mass spec data (Figure 2cd, Supplementary Figure 1 and Table 1), most of RNA fragments including modifications were unequivocally assigned to each individual tRNA. The quality of tRNA isolation is assured. The tRNA isolation method we used here has been established very well, and applied to many projects from my laboratory and collaborators.

- As mentioned above, there is no follow-up on the QTRT1 or QTRT2-knockout (KO) cell lines. The addition of molecular or biological effects caused by lack of Q modification would have greatly strengthened the manuscript and provided insight into the function of the Q modification in mitochondrial tRNAs.

As described above, we here provide mitoribosome profiling (Figure 6c) and protein aggregation assay (Figure 6d) as follow-up experiments for Q34 modification.

REVIEWERS' COMMENTS:

Reviewer #1 (Remarks to the Author):

The authors have addressed all my previous concerns and the revised manuscript has been improved. I thus strongly support the publication in Nature Communications.

Reviewer #2 (Remarks to the Author):

The authors addressed my questions. I have no further questions.

Reviewer #3 (Remarks to the Author):

The Authors have addressed all concerns from the original review. The authors have provided a reason for why a comprehensive map of human tRNA modifications is important since bovine and human mt-tRNAs differ fairly significantly in sequence so simple prediction is not sufficient. In addition, the authors provide evidence for new modifications, cmnm5U and cmnm5s2U. The authors also add ribosome profiling data and protein folding reporter plasmids to provide functional relevance for the findings. Based upon these clarifications and additions, the paper has provided novelty and biological insight that would be of broad interest in the field of RNA biology. Thus, I recommend the manuscript for publication.